# Risk-aware Direct Preference Optimization under Nested Risk Measure

**Lijun Zhang[1], Lin Li[1], Yajie Qi[1], Huizhong Song[1], Yaodong Yang[2], Jun Wang[3], Wei Wei[1]***

1. Key Laboratory of Computational Intelligence and Chinese Information Processing of
Ministry of Education, School of Computer and Information Technology,
Shanxi University, Taiyuan, Shanxi, China.
2. Institute for AI, Peking University, Beijing, China.
3. University College London.

## Abstract

When fine-tuning pre-trained Large Language Models (LLMs) to align with human values and intentions, maximizing the estimated reward can lead to superior performance, but it also introduces potential risks due to deviations from the reference model's intended behavior. Most existing methods typically introduce KL divergence to constrain deviations between the trained model and the reference model; however, this may not be sufficient in certain applications that require tight risk control. In this paper, we introduce Risk-aware Direct Preference Optimization (Ra-DPO), a novel approach that incorporates risk-awareness by employing a class of nested risk measures. This approach formulates a constrained risk-aware advantage function maximization problem and converts the Bradley-Terry model into a token-level representation. The objective function maximizes the likelihood of the policy while suppressing the deviation between a trained model and the reference model using a sequential risk ratio, thereby enhancing the model's risk-awareness. Experimental results across three open-source datasets: IMDb Dataset, Anthropic HH Dataset, and AlpacaEval, demonstrate the proposed method's superior performance in balancing alignment performance and model drift.

## 1 Introduction

Learning from human feedback, serving as a bridge to align LLMs with human preferences, is crucial for ensuring that the generations are more helpful, factual, and ethical, among other desiderata [1, 2, 3, 4]. Alignment methods such as RLHF [2, 3] and DPO [5] have consistently proven more effective than supervised finetuning (SFT) alone. Notably, DPO, featuring a simple and straightforward training process, directly uses the likelihood of the policy to define an implicit reward fitted to the preference data, which has emerged as a popular alternative since it bypasses explicit reward modeling challenges while delivering competitive performance. Subsequently, a variety of DPO variants have been proposed, such as f-DPO [6], IPO [7], RDPO [8], and SimPO [9], to enhance performance. However, a key limitation of these methods is that they only consider evaluation at the sentence level, ignoring the fact that the generation of these responses occurs sequentially, following an auto-regressive approach.

Recently, a fresh perspective on LLMs alignment has been introduced, specifically a sequential and token-level direct preference optimization known as TDPO [10]. This method allows for examining divergence in relation to a reference model on a more granular, token-by-token basis. Specifically, inspired by Trust Region Policy Optimization (TRPO) [11] from reinforcement learning (RL) field [12, 13], TDPO redefines the objective of maximizing restricted rewards in a sequential manner and

---

*Correspondence to <`weiwei@sxu.edu.cn`>.

bridges sentence-level reward to token-level generation through the Bellman equation. However, since the objective at each step is to maximize the expected return, a risk-neutral criterion, which neglects the characteristics of the reward distribution beyond the mean, TDPO cannot guarantee a low risk of deviation from the reference model during alignment training. This could be catastrophic for practical applications, as a significant deviation from the reference model typically implies the degradation of superior decision-making and reasoning capabilities.

Fortunately, in the field of RL, a series of risk-sensitive methods [14, 15, 16] have been proposed that achieve superior performance by introducing various risk measures. Recently, some researchers have attempted to introduce this technology to align LLMs with human preferences. For instance, RA-RLHF [17] introduces a static risk measure into the fine-tuning of RL, while KTO [18] introduces prospect theory [19] to fit human choice behavior when faced with uncertain events. However, these methods only consider the risk at the sentence level by analyzing the distribution characteristics of the preference data, thereby overlooking the inherently sequential and auto-regressive process of response generation.

In this paper, we focus on the risk in token-level generation when aligning LLMs with human values and intentions. Specifically, from a risk-sensitive perspective, we investigate a novel direct preference optimization method and provide corresponding theoretical and empirical results. Our main contributions are summarized as follows.

- We design a new risk-aware, token-level objective function and prove that maximizing this objective leads to policy improvements. Furthermore, by deriving the mapping from the risk-aware state-action value function to the optimal policy and establishing the equivalence between the Bradley-Terry model and the Regret Preference Model, we obtain an optimization objective that is solely dependent on the risk-sensitive policy.

- We propose a novel Risk-aware Direct Preference Optimization (Ra-DPO) method. The method maintains a natural and simple loss function, specifically, the sum of the DPO loss and the negative sequential risk ratio (see Figure 1). This loss function maximizes policy likelihood while suppressing deviation from reference model through the sequential risk ratio, thereby enhancing risk-awareness in striking a balance between alignment performance and model drift.

- Experimentally, we evaluate the effectiveness of our proposed method across various text generation tasks and assess its sensitivity to the risk control parameter. The experimental results demonstrate that our method can effectively suppress the risk of model drift while enhancing its performance.

## 2 Preliminaries

### 2.1 Preference-based Policy Optimization

Considering a preference-based language model fine-tuning task, let $x$ denote an input prompt (question), and $y$ denote the generated response (answer). The notation $y_w \succ y_l \mid x$ symbolizes the human preference data, where $y_w$ (win) represents a response that is more preferred by humans compared to $y_l$ (lose). Both $x$ and $y_w/y_l$ are sequences of tokens.

**Bradley-Terry Model.** In preference-based fine-tuning, to align with human preferences, a preference predictor adhering to the Bradley-Terry (BT) [20] model has been widely employed for pairwise comparisons. The likelihood of a preference pair is commonly expressed as:

$$P_{\text{BT}}\left(y_w \succ y_l \mid x\right) = \frac{\exp\left(r^*\left(x, y_w\right)\right)}{\exp\left(r^*\left(x, y_w\right)\right) + \exp\left(r^*\left(x, y_l\right)\right)}, \tag{1}$$

where $r^*(x, y_w)$ and $r^*(x, y_l)$ stand for the reward function at the sentence level from the preferred and dispreferred answers, respectively.

**Direct Preference Optimization.** DPO [5] begins with the following RL objective:

$$\max_{\pi_\theta} \mathbb{E}_{x \sim \mathcal{D}}\left[\mathbb{E}_{y \sim \pi_\theta(\cdot|x)}\left[r\left(x, y\right) - \beta D_{\text{KL}}\left(\pi_\theta(\cdot \mid x) \| \pi_{\text{ref}}(\cdot \mid x)\right)\right]\right], \tag{2}$$

where $\mathcal{D}$ represents the human preference dataset, $\beta$ is the coefficient of the reverse KL divergence penalty, $\pi_{\text{ref}}(\cdot \mid x)$ is the policy of a fixed reference model (typically selected to be the model that

has undergone post-supervised fine-tuning), and $\pi_\theta \left( \cdot \mid x \right)$ represents the policy of the trained model, initialized with $\pi_\theta = \pi_{\text{ref}}$.

By reparameterizing the reward function in Equation (2), DPO establishes a direct functional mapping between the reward model and the optimal policy:

$$r(x, y) = \beta \log \frac{\pi_\theta^*(y \mid x)}{\pi_{\text{ref}}(y \mid x)} + \beta \log Z(x), \tag{3}$$

where $Z(x)$ is the partition function. Subsequently, Equation (2) can be reformulated as DPO loss:

$$\mathcal{L}_{\text{DPO}} \left( \pi_\theta; \pi_{\text{ref}} \right) = -\mathbb{E}_{(x, y_w, y_l) \sim \mathcal{D}} \left[ \log \sigma \left( u \left( x, y_w, y_l \right) \right) \right], \tag{4}$$

where $u \left( x, y_w, y_l \right) = \beta \log \frac{\pi_\theta(y_w \mid x)}{\pi_{\text{ref}}(y_w \mid x)} - \beta \log \frac{\pi_\theta(y_l \mid x)}{\pi_{\text{ref}}(y_l \mid x)}$.

## 2.2 Preference-based Markov Decision Process

A Preference-based Markov Decision Process (Pb-MDP) can be formulated as a modification of the classical MDP: $\mathcal{M} = \langle \mathcal{S}, \mathcal{A}, r, \mathbf{P}, \gamma, T \rangle$, where $\mathcal{S}$ and $\mathcal{A}$ represent the finite state and action spaces, respectively; $\mathbf{P} : \mathcal{S} \times \mathcal{A} \to \mathcal{S}$ is the probabilistic transition function; $r$ represents the reward function over the entire prompt-response, which is defined as $(\mathcal{S} \times \mathcal{A})^T \to \mathcal{R}$; $\gamma$ is the discount factor, and $T$ denotes the length of a trajectory or episode.

Specifically, for language generation, the state $s_t = [x, y^{<t}] \in \mathcal{S}$ usually consists of the prompt and the generated response up to the previous step, and action $a_t = y^t \in \mathcal{A}$ corresponds to the current generated token. Additionally, note that $y^{<1} = [\ ]$ is an empty sequence. Therefore, we denote $[x] = [x, [\ ]] = [x, y^{<1}]$. For a given prompt $x$ and the first $t-1$ tokens $y^{<t}$ of the response $y$, the probability distribution of the next token conditioned on $[x, y^{<t}]$ is denoted by $\pi_\theta(\cdot \mid [x, y^{<t}])$.

## 2.3 Risk Measure

It is more desirable to keep risk under control for language generation tasks rather than relying solely on a risk-neutral criterion, which ignores the distributional characteristics of rewards, especially in applications that may have potential broad societal impact. Therefore, we introduce the risk-sensitive criterion [21, 22] to quantify potential hidden risks. More specifically, we provide an introduction to the risk-sensitive function and nested risk measure as follows.

**Risk-sensitive Function.** In this paper, the risk-sensitive function is required to satisfy the following properties for all $Z, Z' \in \mathcal{Z}$: *Concavity:* $\forall \ \lambda \in [0, 1] : \eta \left( \lambda Z + (1 - \lambda) Z' \right) \geq \lambda \eta \left( Z \right) + (1 - \lambda) \eta \left( Z' \right)$; *Translation Invariance:* $\forall \ \epsilon \in \mathbb{R} : \eta \left( Z + \epsilon \right) = \eta \left( Z \right) + \epsilon$. This class captures a broad range of useful objectives, including the popular Conditional Value-at-Risk (CVaR) [23, 24, 25] and Entropic Risk Measure (ERM) [26, 27].

**Nested Risk-measures.** In the context of standard Pb-MDP, nested risk measures [28, 29, 30] can be expressed in Bellman equation type as follows:

$$\begin{cases} Q_\pi \left( [x, y^{<t}], y^t \right) = R \left( [x, y^{<t}], y^t \right) + \Phi^\mu \left( V_\pi \left( [x, y^{<t+1}] \right) \right), \\ V_\pi \left( [x, y^{<t}] \right) = \mathbb{E}_\pi \left[ Q_\pi \left( [x, y^{<t}], y^t \right) \right], \\ V_\pi \left( [x, y^{<T}] \right) = R \left( [x, y^{<T}] \right), \end{cases} \tag{5}$$

where $\Phi(\cdot)$ is a risk measure function with a risk control parameter $\mu$, $Q_\pi \left( [x, y^{<t}], y^t \right)$ and $V_\pi \left( [x, y^{<t}] \right)$ represent the state-action value and state value under the nested risk measures at timestep $t \in [1, \cdots, T]$, respectively.

Due to space constraints, we provide a detailed survey on risk measures in Appendix A.1 and the expanded version of the value function definition in Appendix A.2.

## 3 Methodology

This section proposes a novel language model alignment method named Risk-aware Direct Preference Optimization (Ra-DPO). Specifically, we first analyze the characteristics of nested risk measures and design a new risk-aware token-level objective function by reformulating the constrained reward

maximization problem into a token-level form. Subsequently, we prove that maximizing the objective function leads to policy improvements. Then, an optimization objective solely related to the risk-sensitive policy is obtained by deriving the mapping from the risk-aware state-action function to the optimal policy and establishing BT model equivalence with the Regret Preference Model. Finally, we conduct a formal analysis of this optimization objective in terms of derivatives and derive the loss function for Ra-DPO.

## 3.1 Risk-aware Objective Function

In this subsection, we aim to design a new risk-aware objective function for preference-based language model fine-tuning. Unfortunately, although the recursive Bellman equation under nested risk measures was introduced in Subsection 2.3, it cannot be directly applied due to the following reasons: (1) For the Pb-MDP setting, the algorithm can only obtain the reward (an implicit reward fitted to the preference data) over the entire prompt-response and thus cannot compute the target value at each step. (2) The nested risk-measures incorporate a Bellman-type recursion and are not law-invariant [31], making them complex and difficult to compute.

To surmount these obstacles, a straightforward approach is to introduce the state augmentation method, that is, to reconstruct an augmented Pb-MDP [30], where the state at each timestep includes a prompt $x$ and the first $t-1$ tokens $y^{<t}$ of the response. This approach has the property that the state at the previous timestep is a subset of the state at the current timestep, i.e., $[x, y^{<t-1}] \subset [x, y^{<t}]$. This approach can reformulate the recursive Bellman equation into a classical Bellman equation while satisfying the standard requirements for transformer-based long-sequence modeling in LLMs. Therefore, in this paper, we directly define the state as a combination of the prompt and the generated response up to the current step to model the sequential and auto-regressive generation. Then, the nested risk-aware objective's Bellman equation in Equation (5) can be rewritten as:

$$
\begin{cases}
\tilde{Q}_\pi \left( [x, y^{<t}], y^t \right) = \Phi^\mu \left( \tilde{V}_\pi \left( y^{t+1} \circ ([x, y^{<t}], y^t) \right) \right), \\
\tilde{V}_\pi \left( [x, y^{<t}] \right) = \mathbb{E}_\pi \left[ \tilde{Q}_\pi \left( [x, y^{<t}], y^t \right) \right], \\
\tilde{V}_\pi \left( [x, y^{<T}] \right) = R \left( [x, y^{<T}] \right),
\end{cases}
\tag{6}
$$

where $\tilde{Q}_\pi \left( [x, y^{<t}], y^t \right)$ and $\tilde{V}_\pi \left( [x, y^{<t}] \right)$ represent the risk-aware state-action value and state value under the policy $\pi$, respectively. The operator $\circ$ denotes the concatenation of the state and action.

It is noteworthy that there is a significant difference in the calculation of $\tilde{V}_\pi \left( [x, y^{<t}] \right)$ and $V_\pi \left( [x, y^{<t}] \right)$. According to Lemma 3.6 in [30], we can obtain the following lemma, whose proof is provided in Appendix B.1.

**Lemma 3.1.** *For a given Pb-MDP, the reward over the entire prompt-response can be decomposed as $r = \sum_{t=1}^{T} \gamma^{t-1} R([x, y^{<t}], y^t)$, the relationship between the state value function Equation (5) and Equation (6) is as follows: $\tilde{V}_\pi \left( [x, y^{<t}] \right) = V_\pi \left( [x, y^{<t}] \right) + R_{1:t-1}$, where $R_{1:t-1} = \sum_{h=1}^{t-1} \gamma^{h-1} R \left( [x, y^{<h}], y^h \right)$ denotes the cumulative reward of the $1 \sim t-1$ steps of the prompt-response, and $V_\pi[x]$ and $\tilde{V}_\pi[x]$ are equivalent.*

Subsequently, based on Equation (6), we define the risk-aware advantage function as follows.

**Definition 3.2.** For a risk-sensitive Pb-MDP that satisfies the Bellman equation in Equation (6), the risk-aware advantage function can be defined as:

$$
\tilde{A}_\pi \left( [x, y^{<t}], z \right) = \tilde{Q}_\pi \left( [x, y^{<t}], z \right) - \Phi^\mu(\tilde{V}_\pi \left( [x, y^{<t}] \right)),
\tag{7}
$$

where $z \sim \pi_\theta \left( \cdot \mid [x, y^{<t}] \right)$.

The definition is reasonable: its derivation is provided in Appendix B.2. Furthermore, based on the definition of risk-aware advantage function in Definition 3.2, we propose a new risk-aware objective function:

$$
\max_{\pi_\theta} \mathbb{E}_{x, y^{<t} \sim \mathcal{D}, z \sim \pi_\theta(\cdot|[x, y^{<t}])} \left[ \tilde{A}_{\pi_{\text{ref}}} \left( [x, y^{<t}], z \right) - \beta D_{\text{KL}} \left( \pi_\theta \left( \cdot \mid [x, y^{<t}] \right) \| \pi_{\text{ref}} \left( \cdot \mid [x, y^{<t}] \right) \right) \right].
\tag{8}
$$

The objective function maximizes a risk-sensitive advantage function subject to a KL divergence constraint, which accounts for risk during selecting the policy, thereby striking a better balance

between alignment performance and model drift. It is worth emphasizing that maximizing the risk-aware objective function in Equation (8) leads to policy improvements, as stated in the following lemma, whose proof is provided in Appendix B.3.

**Lemma 3.3.** *Given two policies $\pi$ and $\pi'$, if for any state $s_t = [x, y^{<t}]$, $\mathbb{E}_{z \sim \pi'}\left[\tilde{A}_\pi\left([x, y^{<t}], z\right)\right] \geq 0$, then we can conclude: $\mathbb{E}_{x \sim \mathcal{D}}\left[\tilde{V}_{\pi'}([x])\right] \geq \mathbb{E}_{x \sim \mathcal{D}}\left[\tilde{V}_\pi([x])\right]$.*

## 3.2 Risk-aware Preference Optimization

In this subsection, we convert the BT model into risk-sensitive token-level representation, which is divided into two steps: (1) derive the mapping from the risk-aware state-action function to the optimal policy; (2) establish the equivalence between the BT model and the Regret Preference Model.

Specifically, starting from Equation (8), the mapping from the risk-aware state-action function $\tilde{Q}_\pi$ to the optimal policy $\pi_\theta^*$ can be derived as stated in the following lemma.

**Lemma 3.4.** *The constrained problem in Equation (8) has the closed-form solution:*

$$\pi_\theta^*\left(z \mid [x, y^{<t}]\right) = \frac{\pi_{\text{ref}}\left(z \mid [x, y^{<t}]\right) \exp\left(\frac{1}{\beta}\tilde{Q}_{\pi_{\text{ref}}}\left([x, y^{<t}], z\right)\right)}{Z\left([x, y^{<t}]; \beta\right)}, \tag{9}$$

*where $Z\left([x, y^{<t}]; \beta\right) = \mathbb{E}_{z \sim \pi_{\text{ref}}(\cdot \mid [x, y^{<t}])} e^{\frac{1}{\beta}\tilde{Q}_{\pi_{\text{ref}}}([x, y^{<t}], z)}$ is the partition function.*

The proof is provided in Appendix B.4. Then, by rearranging Equation (9), we obtain the expression of the risk-aware state-action function in terms of the policy:

$$\tilde{Q}_{\pi_{\text{ref}}}\left([x, y^{<t}], z\right) = \beta \log \frac{\pi_\theta^*\left(z \mid [x, y^{<t}]\right)}{\pi_{\text{ref}}\left(z \mid [x, y^{<t}]\right)} + \beta \log Z\left([x, y^{<t}]; \beta\right). \tag{10}$$

Subsequently, by utilizing the reward decomposition formula $r = \sum_{t=1}^{T} \gamma^{t-1} R\left([x, y^{<t}], y^t\right)$ from Lemma 3.1, we establish BT model equivalence with the Regret Preference Model as shown in the following lemma, whose proof is provided in Appendix B.5.

**Lemma 3.5.** *Given a reward function $r(x, y)$ of the entire prompt-response, based on the relationship between the token-wise rewards and the reward function $r(x, y) = \sum_{t=1}^{T} \gamma^{t-1} R\left([x, y^{<t}], y^t\right)$, we can establish the equivalence between the Bradley-Terry model and the Regret Preference Model, i.e.,*

$$P_{\text{BT}}\left(y_1 \succ y_2 \mid x\right) = \sigma\left(\sum_{t=1}^{T_1} \gamma^{t-1} \tilde{A}_\pi\left([x, y_1^{<t}], y_1^t\right) - \sum_{t=1}^{T_2} \gamma^{t-1} \tilde{A}_\pi\left([x, y_2^{<t}], y_2^t\right)\right), \tag{11}$$

*where $\sigma(z) = 1/(1 + \exp(-z))$ is the logistic sigmoid function for any random variable $z$.*

According to the definition of the risk-aware advantage function in Definition 3.2, we can directly establish the relationship between the optimal solution in Equation (10) and preference optimization objective in Equation (11). In this way, we reformulate the BT model to be directly tied to the risk-aware optimal policy $\pi_\theta^*$ and the reference policy $\pi_{\text{ref}}$, which is summarized in the following theorem, whose proof is provided in the Appendix B.6.

**Theorem 3.6.** *Given prompts $x$ and pairwise responses $(y_1, y_2)$, and the risk-aware objective function in Equation (8), the Bradley-Terry model expresses the human preference probability in terms of the risk-aware optimal policy $\pi_\theta^*$ and reference policy $\pi_{\text{ref}}$:*

$$P_{\text{BT}}^*\left(y_1 \succ y_2 \mid x\right) = \sigma\left(u^*\left(x, y_1, y_2\right) - \delta^*\left(x, y_1, y_2\right)\right), \tag{12}$$

*where $u\left(x, y_1, y_2\right)$ represents the difference in implicit rewards defined by the risk-aware policy $\pi_\theta^*$ and the reference policy $\pi_{\text{ref}}$, weighted by $\beta$, represented as:*

$$u\left(x, y_1, y_2\right) = \beta \log \frac{\pi_\theta\left(y_1 \mid x\right)}{\pi_{\text{ref}}\left(y_1 \mid x\right)} - \beta \log \frac{\pi_\theta\left(y_2 \mid x\right)}{\pi_{\text{ref}}\left(y_2 \mid x\right)}, \tag{13}$$

*and $\delta\left(x, y_1, y_2\right)$ represents the difference in sequential risk ratios between two pairs $(x, y_1)$ and $(x, y_2)$, expressed as:*

$$\delta\left(x, y_1, y_2\right) = \beta D_{\text{SeqRR}}\left(x, y_2; \pi_{\text{ref}} \mid \pi_\theta\right) - \beta D_{\text{SeqRR}}\left(x, y_1; \pi_{\text{ref}} \mid \pi_\theta\right), \tag{14}$$

*where $D_{\text{SeqRR}}\left(x, y; \pi_{\text{ref}} \mid \pi_\theta\right) = \sum_{t=1}^{T} \Phi_{z \sim \pi_{\text{ref}}}^\mu\left(\log \frac{\pi_{\text{ref}}(z \mid [x, y^{<t}])}{\pi_\theta(z \mid [x, y^{<t}])}\right)$.*

## 3.3 Loss Function and Formal Analysis

Drawing on Theorem 3.6, we reformulate the BT model into a structure solely relevant to the risk-sensitive policy, which enables us to formulate a likelihood maximization objective for a parametrized policy $\pi_\theta$. The loss function is given by:

$$\mathcal{L}_{\text{Ra-DPO}_1}(\pi_\theta; \pi_{\text{ref}}) = -\mathbb{E}_{(x, y_w, y_l) \sim \mathcal{D}}[\log \sigma(u(x, y_w, y_l) - \delta(x, y_w, y_l))]. \tag{15}$$

In Equation (15), the sequential risk ratio is explicitly introduced into the loss function, which incorporates risk-awareness to balance alignment performance and model drift. To elucidate the benefits of the proposed method, we conduct the further analysis of the loss function and its corresponding gradient. For brevity, we use $u$ to denote $u(x, y_w, y_l)$, and $\delta$ to represent $\delta(x, y_w, y_l)$. By simple calculations, we can derive the gradient of loss function in Equation (15) with respect to parameter $\theta$ :

$$\nabla_\theta \mathcal{L}_{\text{Ra-DPO}_1}(\pi_\theta; \pi_{\text{ref}}) = -\mathbb{E}_{(x, y_w, y_l) \sim \mathcal{D}}[(-u + \delta)[\nabla_\theta u - \nabla_\theta \delta]], \tag{16}$$

where $(-u + \delta)$ serves as the weighting factor for the gradient.

From Equation (16), we can observe that the first part, $(-u)$, corresponds to the weight factor in the first part of loss function of TDPO. Its value increases when the language model makes prediction errors relative to human preferences, i.e., $\log \frac{\pi_\theta(y_l|x)}{\pi_{\text{ref}}(y_l|x)} > \log \frac{\pi_\theta(y_w|x)}{\pi_{\text{ref}}(y_w|x)}$. The second part, $\delta$, consists of the difference between the sequential risk ratios of the dispreferred and preferred response subsets, which is a distinctive component of our method. When selecting a convex function (risk-averse), such as CVaR, as the risk measure, our method can automatically control the risk ratio balance.

Furthermore, building upon a common objective shared by our method and TDPO [10], i.e., reducing risks stemming from model drift and ensuring training stability, we further provide a second version of our method, Ra-DPO$_2$. The loss function of Ra-DPO$_2$ is given by:

$$\mathcal{L}_{\text{Ra-DPO}_2}(\pi_\theta; \pi_{\text{ref}}) = -\mathbb{E}_{(x, y_w, y_l) \sim \mathcal{D}}[\log \sigma(u(x, y_w, y_l) - \alpha \delta_2(x, y_w, y_l))], \tag{17}$$

where $\delta_2(x, y_1, y_2) = \beta D_{\text{SeqRR}}(x, y_2; \pi_{\text{ref}} \mid \pi_\theta) - \text{sg}(\beta D_{\text{SeqRR}}(x, y_1; \pi_{\text{ref}} \mid \pi_\theta))$. The operator sg represents the stop-gradient operator, which blocks the propagation of gradients. The parameter $\beta$ can control the deviation between $D_{\text{SeqRR}}(x, y_2; \pi_{\text{ref}} \mid \pi_\theta)$ and $(\beta D_{\text{SeqRR}}(x, y_1; \pi_{\text{ref}} \mid \pi_\theta))$. Ra-DPO$_2$ modifies the loss function of Ra-DPO$_1$ by disabling the gradient propagation of $D_{\text{SeqRR}}(x, y_w; \pi_{\text{ref}} \mid \pi_\theta)$ and treating it as a baseline term for alignment of $D_{\text{SeqRR}}(x, y_l; \pi_{\text{ref}} \mid \pi_\theta)$. The aim of the modification is to ensure training stability, rather than to accelerate training speeding. To summarize, the comparison of the loss functions for DPO, TDPO$_2$, and Ra-DPO$_2$ is shown in Figure 1. In addition, we provide a procedure of our method, and provide its pseudocode (Algorithm 1) in Appendix B.7.

$$\mathcal{L}_{\text{DPO}}(\pi_\theta; \pi_{\text{ref}}) = -\mathbb{E}\left[\log \sigma\left(\beta \log \frac{\pi_\theta(y_w \mid x)}{\pi_{\text{ref}}(y_w \mid x)} - \beta \log \frac{\pi_\theta(y_l \mid x)}{\pi_{\text{ref}}(y_l \mid x)}\right)\right]$$

$$\mathcal{L}_{\text{TDPO}_2}(\pi_\theta; \pi_{\text{ref}}) = -\mathbb{E}\left[\log \sigma\left(\left(\beta \log \frac{\pi_\theta(y_w \mid x)}{\pi_{\text{ref}}(y_w \mid x)} - \beta \log \frac{\pi_\theta(y_l \mid x)}{\pi_{\text{ref}}(y_l \mid x)}\right) - \alpha\left(\beta D_{\text{SeqKL}}(x, y_l; \pi_{\text{ref}} \| \pi_\theta) - \text{sg}(\beta D_{\text{SeqKL}}(x, y_w; \pi_{\text{ref}} \| \pi_\theta))\right)\right)\right]$$

$$\mathcal{L}_{\text{Ra-TDPO}_2}(\pi_\theta; \pi_{\text{ref}}) = -\mathbb{E}\left[\log \sigma\left(\left(\beta \log \frac{\pi_\theta(y_w \mid x)}{\pi_{\text{ref}}(y_w \mid x)} - \beta \log \frac{\pi_\theta(y_l \mid x)}{\pi_{\text{ref}}(y_l \mid x)}\right) - \alpha\left(\beta D_{\text{SeqRR}}(x, y_l; \pi_{\text{ref}} \| \pi_\theta) - \text{sg}(\beta D_{\text{SeqRR}}(x, y_w; \pi_{\text{ref}} \| \pi_\theta))\right)\right)\right]$$

Figure 1: Comparison of loss functions for DPO, TDPO$_2$ and Ra-DPO$_2$ methods. The sg denotes the stop-gradient operator.

## 4 Experiments

We empirically evaluate our method on several open-source datasets and pre-trained models, aiming to investigate the following questions: (1) How does the performance of our method compare with that of existing methods, particularly in terms of risk sensitivity when handling challenging text generation tasks? (2) How does the risk control parameter $\mu$ affect the performance of our method?

To answer these questions, we conducted experiments on IMDb Dataset [32], Anthropic HH Dataset [33], and AlpacaEval [34] for three different text generation tasks. Based on the original *KTO implementation*[2], we trained Ra-DPO and baseline models using the same hyperparameters.

---
[2]Available at `https://github.com/ContextualAI/HALOs`

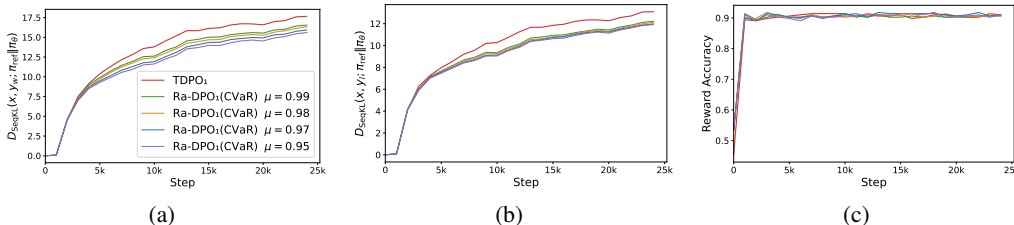

(a)          (b)          (c)

Figure 2: The experiment on the IMDb dataset with GPT-2 Large serving as the base model. (a) and (b) present the progression of sequential KL divergence (the lower the better) for both preferred and dispreferred responses. (c) illustrates the reward accuracy curves (the higher the better).

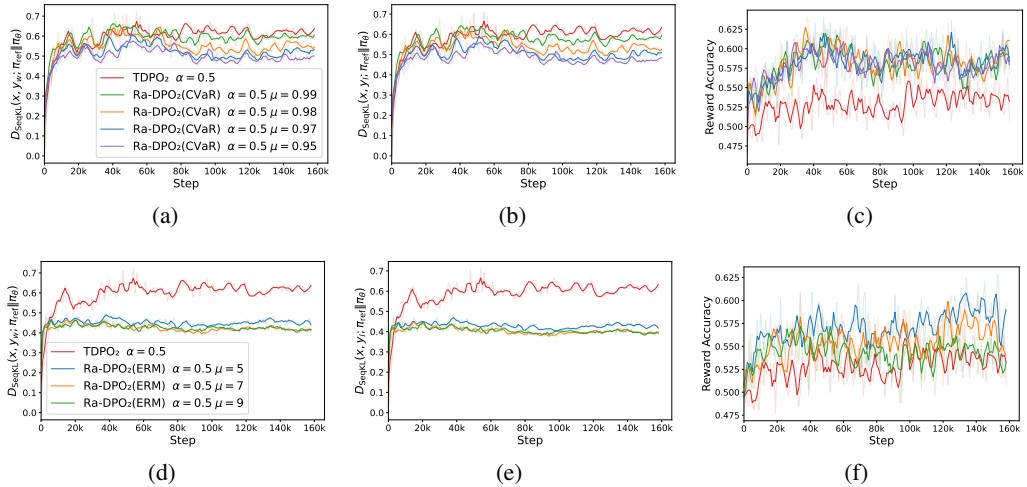

(a)          (b)          (c)

(d)          (e)          (f)

Figure 3: The experiment on the Anthropic HH dataset with Pythia-1.4B serving as the base model. **Left** and **Middle** present the progression of sequential KL divergence (the lower the better) for both preferred and dispreferred responses. **Right** illustrates reward accuracy curves (the higher the better).

Specifically, for Ra-DPO, we employed nested risk measures based on CVaR [24] and ERM [27]. We compare our method against the following algorithms: (1) DPO [5], which considers evaluation at the sentence level; (2) PPO [35], an offline PPO variant provided by the original KTO implementation; (3) $TDPO_1$ and $TDPO_2$ [10], which convert the BT model into token-level representations; (4) KTO [18], which considers preferences in human decisions that are not aimed at maximizing utility. Experimental setup and results are reported in Subsections 4.1-4.3 and Appendix C.

## 4.1 Experiments on IMDb Dataset

**Experimental Setup:** The IMDb dataset is a controlled semantic generation dataset within the context of movie reviews, serving as a valuable resource for training and evaluating sentiment analysis models. We employ GPT-2 Large [36] as the base model and use the model checkpoint *insub/gpt2-large-IMDb-fine-tuned*[3] as the SFT model. The results of the versions of $Ra$-$DPO_1$ (CVaR) with risk control parameter $\mu \in \{0.99, 0.98, 0.97, 0.95\}$ are shown in Figure 2.

**Evaluation:** Figure 2 shows that $Ra$-$DPO_1$ can outperform or achieve reward accuracy similar to the advanced TDPO algorithm while reducing model drift (i.e., lower sequential KL divergence), demonstrating the risk-awareness of $Ra$-$DPO_1$ in balancing alignment performance and model drift.

## 4.2 Experiments on Anthropic HH Dataset

**Experimental Setup:** Anthropic HH dataset contains 170k dialogues between a human and an automated assistant, where each transcript ends with a pair of responses generated by an LLM along with a preference label denoting the human-preferred response. We use Pythia-1.4B and Pythia-2.8B [37] as the base models to test our method on Anthropic HH dataset, respectively. The

---

[3]`https://huggingface.co/insub/gpt2-large-IMDb-fine-tuned`

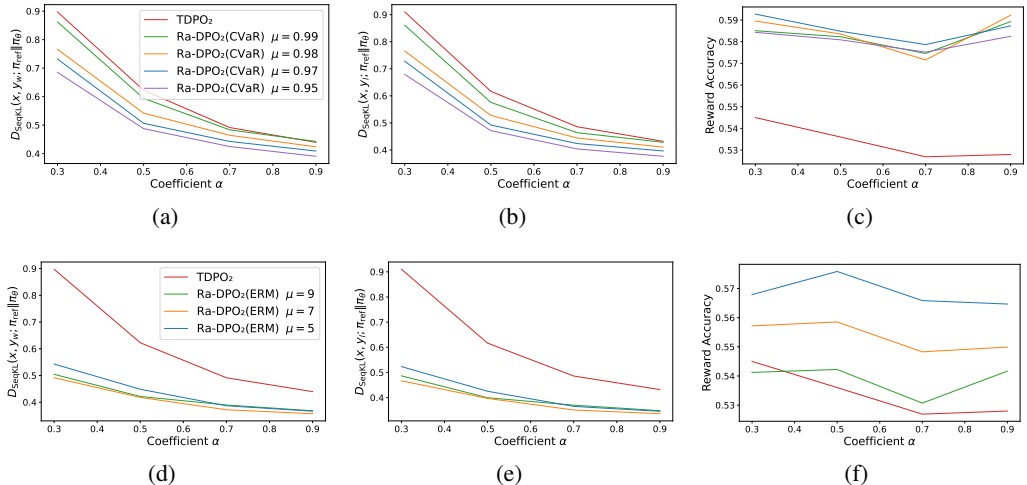

(a)       (b)       (c)

(d)       (e)       (f)

Figure 4: The experiment on the Anthropic HH dataset with Pythia-1.4B serving as the base model. **Left** and **Middle** presents the sequential KL divergence (the lower the better) for preferred and dispreferred responses, while **Right** presents the reward accuracy curves (the higher the better) under $\alpha = \{0.3, 0.5, 0.7, 0.9\}$.

reference models are trained by fine-tuning the base models on chosen completions. The results are depicted in Figure 3, Figure 4, and Appendix C.4.

**Evaluation:** Figure 3 shows the performance of $TDPO_2$ and different versions of $Ra\text{-}DPO_2$ with respect to the risk control parameter $\mu$ while keeping coefficient $\alpha$ constant at 0.5. Figure 4 presents the statistical results of different algorithms with coefficient $\alpha = \{0.3, 0.5, 0.7, 0.9\}$, and the corresponding curve plots are provided in Appendix C.4. From Figures 3 and 4, we can observe that $Ra\text{-}DPO_2$ almost always achieves superior performance (higher reward accuracy) and maintains minimal model drift (lower sequential KL divergence), under both CVaR-based and ERM-based nested risk measures. Additionally, Figure 4 illustrates that $Ra\text{-}DPO_2$ is highly effective in suppressing deviation from reference model, particularly when $\alpha$ takes on smaller values. We hypothesize that this phenomenon may be attributed to the fact that the sequential risk ratio accumulates step-wise risks through nested risk measures, enabling it to remain responsive to significant model biases even when $\delta_2(x, y_1, y_2)$ in Equation (17) carries a relatively low weight.

### 4.3 Experiments on AlpacaEval

**Experimental Setup:** To comprehensively evaluate $Ra\text{-}DPO_2$ in terms of generation quality, we conducted pairwise comparisons on AlpacaEval using models trained on the Anthropic HH dataset. Following the official *AlpacaEval implementation*[4], we sampled responses with a temperature coefficient of 0.7. The winrate comparisons based on *oasst_pythia_12b*[5] are summarized in Table 1 and Appendix C.4. Both winrate and length-controlled winrate (Lc winrate) are evaluated based on *oasst_pythia_12b*.

Table 1: The compare between different Algorithms and *gpt4_1106_preview*.

| Method | Winrate | Lc winrate |
|---|---|---|
| DPO | 51.1± 1.9 | 44.7± 0.4 |
| PPO | 52.1± 1.8 | 51.9± 0.5 |
| KTO | 51.5± 1.8 | 50.2± 0.6 |
| $TDPO_1$ | 51.9± 1.8 | 53.0± 0.6 |
| $TDPO_2$ | 52.2± 1.6 | 52.2± 0.5 |
| $Ra\text{-}DPO_1$ | **53.5± 1.8** | 53.9± 0.5 |
| $Ra\text{-}DPO_2$ | 52.1± 1.8 | **55.7± 0.5** |

**Evaluation:** Table 1 reveals that under the two indicators of winrate and length-controlled winrate, most of the implemented algorithms can outperform the common default baseline *gpt4_1106_preview* (DPO is more prone to generating long responses). Among them, $Ra\text{-}DPO_1$ and $Ra\text{-}DPO_2$ demonstrate the highest level of performance, especially when it comes to the length-controlled winrate indicator.

---

[4]https://github.com/tatsu-lab/alpaca_eval
[5]https://huggingface.co/OpenAssistant/oasst-sft-4-pythia-12b-epoch-3.5

# 5 Related Work

## 5.1 LLMs Alignment

With the development of LLMs, numerous researchers have encountered challenges stemming from the misaligned next-token prediction task used in the pre-training stage [33, 38, 39, 40], particularly in balancing adherence to human instructions (explicit objectives) with the pursuit of being helpful, honest, and harmless (implicit objectives). Therefore, a typical post-training stage, referred to as preference optimization, is commonly performed to align pre-trained language models with human intentions, and has become an indispensable aspect in the fine-tuning of LLMs. Most approaches [41, 6, 9] only utilize KL divergence at the sentence level to limit significant deviations from the reference model. However, the generation of responses occurs sequentially, following an auto-regressive approach. Recent works [10, 42] introduce a fresh perspective, specifically the token-level direct preference optimization, which allows for examining sequential KL divergence in relation to a reference LLM. However, due to their neglect of reward distribution characteristics other than the mean, these methods suffer from the trouble of being insensitive to risk.

## 5.2 Risk-aware Reinforcement Learning

RL has made groundbreaking achievements [12, 2, 43, 44] through approaches such as Q-learning [45] and policy gradients [11, 35] in sequential decision tasks, but it also faces challenges when applied in the real world [22, 46, 47]. A primary reason is that the risk-neutral criterion (maximizing the expectation) ignores the characteristics of a reward distribution other than the mean, which may be important for certain systems, especially in applications requiring tight risk control [28, 15]. In order to tackle this challenge, two types of risk measures have been introduced: nested and static risk measures. Static risk measures [48, 49, 50] are straightforward to interpret, but the resulting optimal policy may not remain Markovian and may become history-dependent. Nested risk measures [51, 29, 30] utilize MDPs to ensure risk sensitivity of the value iteration at each step under the current state, resulting in a more conservative approach. In this paper, we prefer nested risk measures because they recursively adhere to the Bellman equation and allow the MDPs to be reconstructed through state augmentation, enabling them to remain Markovian.

## 5.3 Risks in LLMs Alignment

When aligning LLMs with human preferences, there are many factors that may pose risks, primarily encompassing the following three types: (1) There may be conflicts among human preferences [52], or human preferences is inherently affected by contextual choice effects [53], thus introducing uncertainty in the objectives when aligning models with human preferences. (2) Humans do not make decisions by maximizing their expected value for uncertain events; instead, they perceive random variables in a biased but well-defined manner [18, 19]. (3) Many popular methods, such as DPO [5], RDPO [8], and simPO [9], introduce the new risks during the alignment training process because they only consider the mean of reward or utility, which is risk-neutral and does not capture the distribution characteristics of rewards efficiently. In this paper, we focus on the third type of risk.

# 6 Discussion

The core objective of preference optimization is to make models less harmful, more helpful, and more truthful. DPO [5] and SimPO [9] serve as representative examples of reference-based and reference-free preference optimization methods, respectively. Although SimPO not only achieves superior performance but also significantly reduces memory consumption, several studies have also pointed out the following limitations: (1) the lack of a reference model reduces training robustness and necessitates stricter conditions to prevent catastrophic forgetting; (2) SimPO introduces dual parameters, which introduce additional complexity on hyperparameter tuning. Therefore, a comprehensive comparison in terms of performance, stability and robustness, hyperparameter tuning complexity, and computational efficiency reveals that each approach has its own trade-offs. Here, we would like to emphasize:

- For preference-based language model fine-tuning task, a trade-off between alignment objectives and model fidelity is still necessary, although the original(reference) model may not be "safe" or "correct". For example, in LLMs safety alignment tasks, a simple objective is to

enable the model to reject unsafe responses while preserving original reasoning capabilities [39, 54]. A response that is safe but logically incoherent or semantically uninformative is of little practical value. Therefore, many studies [41, 11] typically formulate such tasks as constrained reward maximization problems.

- KL divergence has typically been used to penalize excessive deviations from a reference (critic) model [55, 56]. In fact, numerous studies [6, 57] have reported that KL constraint offers many beneficial effects, such as balancing exploration and exploitation, ensuring stability and robustness, preventing catastrophic forgetting, and preserving the model's fundamental capabilities.

# 7 Conclusion

A pressing challenge arises for language generation tasks in the area of risk control, as the models, once trained, are often required to interact directly with humans. In this paper, we propose a novel direct preference optimization method that incorporates risk awareness by introducing nested risk measures into the Bellman equation, to align pre-trained LLMs with human preferences. Specifically, we design a new risk-aware token-level objective function by reformulating the constrained reward maximization problem into a token-level form and then prove that maximizing this objective function leads to improvements in policy performance. Then, an optimization objective solely related to the risk-sensitive policy is obtained by deriving the mapping between the risk-aware state-action function and the optimal policy and establishing BT model equivalence with the Regret Preference Model. Finally, we conduct a formal analysis of this optimization objective and derive the loss function of Ra-DPO, which has practical implications for language generation tasks.

# 8 The Discussion of Limitations and Impacts

## 8.1 Limitations

This paper focuses on the risks associated with token-level generation when aligning LLMs with human values and intentions. Our main contributions include theoretical analysis (see Section 3 and Appendix B), practical algorithm (see Appendix B.7) and simulation verification (see Section 4 and Appendix C). These results characterize the performance of the proposed Ra-DPO in terms of reward accuracy and sequential KL divergence. Below, we discuss the limitations of Ra-DPO from both theoretical and experimental viewpoints.

**Theoretical Viewpoint:** Our theoretical results are based on a class of risk-sensitive functionals that satisfy concavity, monotonicity and translation invariance for any random variables $Z, Z' \in \mathcal{Z}$. Concavity implies risk aversion; translation invariance is an important condition for the validity of Lemma 3.1. Our conclusions may not be valid when such assumptions do not hold. Fortunately, this class captures a broad range of useful objectives, including the popular CVaR [23] and ERM [27].

**Experimental Viewpoint:** Ra-DPO may not be fully effective for tasks such as harmful content moderation and toxicity detection. This is because our primary goal is to reduce the risk of impaired decision-making and reasoning capabilities due to model deviation from the reference model during LLM alignment. However, it is worth noting that the problem we are addressing is both widespread and of significant importance. Furthermore, we recommend a safe or low-risk approach that incorporates risk-awareness within the Safe RLHF [39] or SACPO [54] framework. These approaches explicitly or implicitly model both cost and reward functions while accounting for cost distributions. It may require more computational resources due to the need to train additional models. However, our method can serve as a solid foundation for such potential approaches.

## 8.2 Impact Statement

This paper presents work aimed at making LLMs more helpful. Specifically, we focus on how to reduce the risk of impaired decision-making and reasoning capabilities due to model deviation from the reference model during LLM alignment. Our work has many positive societal impacts, such as providing a theoretical foundation for risk-aware language generation task, none of which we feel must be specifically highlighted. There are no negative societal impacts on our work.

## Acknowledgements

This work were supported in part by the National Natural Science Foundation of China (No.62276160, No.62376013, No.62506221), in part by the Basic Research Program of Shanxi Province (No.202203021211294), and the Shanxi Provincial Overseas Study Fund Project (No.20240002).

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

# A Supplementary Materials for Section 2

## A.1 Risk Measure: A Brief Overview

For quantifying and managing risks, three main paradigms [21, 22, 15] have been developed: the risk-neutral paradigm, the worst-case (i.e., robust) paradigm, and the risk-averse paradigm. The risk-neutral paradigm aims to find a policy that maximize the expected cumulative reward, but it ignores characteristics of the reward distribution other than the mean, which can be crucial for systems with safety concerns. For example, a system may need to operate in a way that mitigates harmful consequences, even in rare and unpredictable situations. The worst-case paradigm [58, 59] focuses on finding a policy that satisfies the constraints of a specific cost function, generally assuming that the maximum possible cost can quantify bounded adversarial disturbances. However, since the worst-case approach assumes disturbances are bounded, it may not work well when those bounds are hard to determine.

The risk-averse paradigm [23, 21, 15], an intermediary paradigm between the risk-neutral and worst-case paradigms, has garnered extensive attention. It describes individuals or algorithms that prefer outcomes with reduced uncertainty by seeking to optimize risk metrics of the possible cumulative reward, emphasizing its distributional characteristics. In general, there are mainly two types of risk measures: nested and static risk-aware measures, each possessing distinct advantages and limitations. Static risk measures [48, 49, 50] are straightforward to interpret, but the resulting optimal policy may not remain Markovian and may become history-dependent. On the other hand, nested risk measures [51, 29, 30] utilize MDPs to ensure risk sensitivity of the value iteration at each step under the current state, resulting in a more conservative approach. We prefer nested risk measures because they recursively adhere to the Bellman equation and allow the MDPs to be reconstructed through state augmentation, thereby enabling them to remain Markovian and ensuring that policy choices depend solely on the current state.

In this paper, we employ a class of nested risk measures, which are variants of the popular CVaR and ERM. Below, we provide introductions to nested risk measures and the CVaR and ERM risk functions.

Specifically, let $(\mathcal{X}, \mathcal{F})$ be a measurable space. A risk measure over $\mathcal{X}$ is a function $\rho : \mathcal{X} \to \mathbb{R}$ that maps uncertain outcomes $X \in \mathcal{X}$ to the real line. A risk measure of the total discounted return $G$ can be described as:

$$\min_{\pi \in \Pi} \rho^\pi(G), \tag{18}$$

where the dependence on $\pi$ emphasizes that the underlying probability measure is induced by the chosen policy. The simplest example is $\rho^\pi = \mathbb{E}^\pi$, for which Equation (18) reduces to the standard risk-neutral RL problem.

**Nested Risk-measures:** Consider a time horizon of length $T \in \mathbb{T}$. A nested risk measure $\rho$ of a random return $G = G_0 + G_1 + \cdots + G_T$ takes the form

$$\rho(G) = \rho_0 \left( G_0 + \rho_1 \left( G_1 + \cdots + \rho_{T-1} \left( G_{T-1} + \rho_T(G_T) \right) \right) \right). \tag{19}$$

where each $\rho_t$ is a risk functional, i.e., a map from a space of random variables to $\mathbb{R} \bigcup \{\infty\}$.

We now introduce the the CVaR and ERM risk functions.

**Conditional value-at-risk (CVaR):** CVaR with risk-aversion level $\alpha \in (0, 1)$ has been defined as:

$$\rho^\pi_{\text{CVaR}}(G; \alpha) = \min_{\eta \in \mathbb{R}} \left\{ \eta + \frac{1}{1 - \alpha} \mathbb{E}^\pi \left[ (G - \eta)^+ \right] \right\},$$

and it has several advantages over VaR: it quantifies the losses encountered in the tail, it can be expressed as a minimization problem, and is a coherent risk measure [60].

**Entropic risk measure (ERM):** ERM is a popular method for measuring risk, defined as:

$$\rho^\pi_{\text{ERM}}(G; \beta) := \frac{1}{\beta} \log \mathbb{E}^\pi [e^{-\beta G}].$$

where $\beta > 0$ indicates the degree of risk aversion. Optimizing ERM is equivalent to optimizing an exponential utility function (EU):

$$\rho^\pi_{\text{EU}}(G; \beta) := \mathbb{E}^\pi [e^{-\beta G}].$$

In this paper, we formulate the nested risk measure within the Preference-based Markov Decision Process (Pb-MDP) framework, and express it in terms of Bellman equation type. To simplify notation, we uniformly denote all such nested risk measures collectively by $\Phi^\mu$, a risk measure with a risk control parameter $\mu$.

## A.2 The Expanded Version of Value Function Definition

The definition of value function for nested risk measure in Equation (5) can be expanded as

$$Q_\pi\left(\left[x, y^{<t}\right], y^t\right) = R\left(\left[x, y^{<t}\right], y^t\right) + \Phi^\mu\left(R\left(\left[x, y^{<t+1}\right], \pi\left(\cdot \mid \left[x, y^{<t+1}\right]\right)\right)\right.$$
$$\left. + \Phi^\mu\left(\cdots \Phi^\mu\left(R\left(\left[x, y^{<T}\right], \pi\left(\cdot \mid \left[x, y^{<T}\right]\right)\right)\right)\right)\right), \tag{20}$$

$$V_\pi\left(\left[x, y^{<t}\right]\right) = R\left(\left[x, y^{<t}\right], \pi\left(\cdot \mid \left[x, y^{<t}\right]\right)\right) + \Phi^\mu\left(R\left(\left[x, y^{<t+1}\right], \pi\left(\cdot \mid \left[x, y^{<t+1}\right]\right)\right)\right.$$
$$\left. + \Phi^\mu\left(\cdots \Phi^\mu\left(R\left(\left[x, y^{<T}\right], \pi\left(\cdot \mid \left[x, y^{<T}\right]\right)\right)\right)\right)\right). \tag{21}$$

Similarly, the definition of the optimal value function, can be expanded as

$$Q_\pi^*\left(\left[x, y^{<t}\right], y^t\right) = \max\left\{R\left(\left[x, y^{<t}\right], y^t\right) + \Phi^\mu\left(R\left(\left[x, y^{<t+1}\right], \pi\left(\cdot \mid \left[x, y^{<t+1}\right]\right)\right)\right.\right.$$
$$\left.\left. + \Phi^\mu\left(\cdots \Phi^\mu\left(R\left(\left[x, y^{<T}\right], \pi\left(\cdot \mid \left[x, y^{<T}\right]\right)\right)\right)\right)\right)\right\}, \tag{22}$$

$$V_\pi^*\left(\left[x, y^{<t}\right]\right) = \max\left\{R\left(\left[x, y^{<t}\right], \pi\left(\cdot \mid \left[x, y^{<t}\right]\right)\right) + \Phi^\mu\left(R\left(\left[x, y^{<t+1}\right], \pi\left(\cdot \mid \left[x, y^{<t+1}\right]\right)\right)\right.\right.$$
$$\left.\left. + \Phi^\mu\left(\cdots \Phi^\mu\left(R\left(\left[x, y^{<T}\right], \pi\left(\cdot \mid \left[x, y^{<T}\right]\right)\right)\right)\right)\right)\right\}. \tag{23}$$

# B   Supplementary Materials for Section 3

## B.1   The Proof of Lemma 3.1

**Lemma 3.1 Restated.** For a given Pb-MDP, the cumulative reward over the entire prompt-response can be decomposed as $r = \sum_{t=1}^T \gamma^{t-1} R\left([x, y^{<t}], y^t\right)$, the relationship between the state value function Equation (5) and Equation (6) is as follows: $\tilde{V}_\pi\left([x, y^{<t}]\right) = V_\pi\left([x, y^{<t}]\right) + R_{1:t-1}$, where $R_{1:t-1} = \sum_{h=1}^{t-1} \gamma^{h-1} R\left([x, y^{<h}], y^h\right)$ denotes the reward of the $1 \sim t-1$ steps of the prompt-response, and $V_\pi[x]$ and $\tilde{V}_\pi[x]$ are equivalent.

*Proof.* First, according to [61, 62, 30], we can reformulate the Pb-MDP as a decision tree-like MDP:

(1) The state transition graph of the Pb-MDP is connected and acyclic;

(2) Each state in the Pb-MDP corresponds to a unique node in the tree;

(3) There is a single root node from which every other node is reachable via a unique path;

(4) The transition probabilities between states follow the Markov property, i.e., the probability of transitioning to any future state depends only on the current state and not on the sequence of events that preceded it.

Formally, let $S$ be the set of states and $p_{ij}$ be the transition probabilities between states $\mathbf{s}_i$ and $\mathbf{s}_j$. For an Pb-MDP with a tree-like structure, the probabilistic transition matrix $P$ is defined such that:

$$p_{ij} > 0 \text{ if there is an edge between } \mathbf{s}_i \text{ and } \mathbf{s}_j \text{ in the tree, and } p_{ij} = 0 \text{ otherwise.} \tag{24}$$

Moreover, for each non-root node $\mathbf{s}_j$, there exists exactly one $\mathbf{s}_i$ such that $p_{ij} > 0$, and $\mathbf{s}_i$ is the unique parent of $\mathbf{s}_j$ in the tree structure.

To differentiate the two value functions, we denote the value from Equation (6) as $\tilde{V}_\pi\left([x, y^{<t}]\right)$ and the value from Equation (5) as $V_\pi\left([x, y^{<t}]\right)$. Since the reward of the entire prompt-response can be decomposed as $r = \sum_{t=1}^T \gamma^{t-1} R\left([x, y^{<t}], y^t\right)$, we have the following relationship:

$$\tilde{V}_\pi\left(\left[x, y^{<t}\right]\right) = V_\pi\left(\left[x, y^{<t}\right]\right) + R_{1:t-1},$$

where $R_{1:t-1} = \sum_{h=1}^{t-1} \gamma^{h-1} R\left(\left[x, y^{<h}\right], y^h\right)$ denotes the reward of the $1 \sim t - 1$ steps of a prompt-response. We prove this relationship by mathematical induction as follows.

**Initial Case.** Using the tree-like Pb-MDP and the initial conditions of the Bellman equation, at the final step $t = T$, we have

$$\tilde{V}_\pi\left(\left[x, y^{<T}\right]\right) = V_\pi\left(\left[x, y^{<T}\right], \pi\left(\cdot \mid \left[x, y^{<t}\right]\right)\right) + R_{1:T-1}$$
$$= V_\pi\left(\left[x, y^{<T}\right]\right) + R_{1:T-1}. \tag{25}$$

**Induction Step.** We now prove that if $\tilde{V}_\pi\left(\left[x, y^{<t+1}\right]\right) = V_\pi\left(\left[x, y^{<t+1}\right]\right) + R_{1:t}$ holds, then $\tilde{V}_\pi\left(\left[x, y^{<t}\right]\right) = V_\pi\left(\left[x, y^{<t}\right]\right) + R_{1:t-1}$ also holds. Since this policy $\pi$ on tree-like Pb-MDP is fixed, it has only one path to arrive $t$-$th$ state $(s_t = [x, y^{<t}])$, denoted as:

$$\Xi_t\left(s_{T,1}\right) = \Xi_h\left(s_{T,2}\right) \quad \forall s_{T,1}, s_{T,2} \in \left\{s_T \mid S_t\left(s_T\right) = \left[x, y^{<t}\right]\right\}.$$

Therefore, $R_{1:t-1}$ is unique.

$$\begin{aligned}
\tilde{V}_\pi\left(\left[x, y^{<t}\right]\right) &= \Phi^\mu\left(V_\pi\left(\left[x, y^{<t+1}\right]\right) + R_{1:t}\right), \\
&= \Phi^\mu\left(V_\pi\left(\left[x, y^{<t+1}\right]\right) + R\left(\left[x, y^{<t}\right], \pi\left(\cdot \mid \left[x, y^{<t}\right]\right)\right) + R_{1:t-1}\right), \\
&= \Phi^\mu\left(V_\pi\left(\left[x, y^{<t+1}\right]\right) + R\left(\left[x, y^{<t}\right], \pi\left(\cdot \mid \left[x, y^{<t}\right]\right)\right)\right) + R_{1:t-1}, \\
&= V_\pi\left(\left[x, y^{<t+1}\right]\right) + R_{1:t-1},
\end{aligned} \tag{26}$$

where the third equality holds because the risk measure function $\Phi$ satisfies translation invariance. Then, by applying this conclusion, we observe that when $t = 1$, $\tilde{V}_\pi[x] = V_\pi[x]$ holds. Thus, we have proven that for the Pb-MDP, the reward over the entire prompt-response can be decomposed as $r = \sum_{t=1}^{T} \gamma^{t-1} R\left(\left[x, y^{<t}\right], y^t\right)$, and $V_\pi[x]$ in Equation (5) and $\tilde{V}_\pi[x]$ in Equation (6) are equivalent. $\square$

## B.2 The Derivation of Definition 3.2

**Definition 3.2 Restated.** For a risk-sensitive Pb-MDP that satisfies the Bellman equation in Equation (6), the risk-aware advantage function can be defined as

$$\tilde{A}_\pi\left(\left[x, y^{<t}\right], z\right) = \tilde{Q}_\pi\left(\left[x, y^{<t}\right], z\right) - \Phi^\mu(\tilde{V}_\pi\left(\left[x, y^{<t}\right]\right)),$$

where $z \sim \pi_\theta\left(\cdot \mid \left[x, y^{<t}\right]\right)$.

**The Derivation.** In terms of designing the objective function at the token level, TDPO [10] provides us with a valuable insight by introducing the advantage function from the TRPO algorithm in RL field as the target for each step. Building upon TDPO, we consider the risk associated with language generation at each step and devise a novel risk-sensitive advantage function. First, based on assumption that $r = \sum_{t=1}^{T} \gamma^{t-1} R\left(\left[x, y^{<t}\right], y^t\right)$, we can get:

$$\begin{aligned}
r &= \sum_{t=1}^{T} \gamma^{t-1} R\left(\left[x, y^{<t}\right], y^t\right) \\
&= \sum_{t=1}^{T} \gamma^{t-1}\left(R\left(\left[x, y^{<t}\right], y^t\right) + \gamma\,\Phi^\mu\left(\tilde{V}_\pi\left(\left[x, y^{<t+1}\right]\right)\right) - \gamma\,\Phi^\mu\left(\tilde{V}_\pi\left(\left[x, y^{<t+1}\right]\right)\right)\right) \\
&= \Phi^\mu\left(\tilde{V}_\pi\left(\left[x\right]\right)\right) + \sum_{t=1}^{T} \gamma^{t-1}\left(R\left(\left[x, y^{<t}\right], y^t\right) + \gamma\,\Phi^\mu\left(\tilde{V}_\pi\left(\left[x, y^{<t+1}\right]\right)\right)\right. \\
&\quad \left. - \Phi^\mu\left(\tilde{V}_\pi\left(\left[x, y^{<t}\right]\right)\right)\right) - \gamma^T\,\Phi^\mu\left(\tilde{V}_\pi\left(\left[x, y^{<T+1}\right]\right)\right) \\
&= \Phi^\mu\left(\tilde{V}_\pi\left(\left[x\right]\right)\right) + \sum_{t=1}^{T} \gamma^{t-1}\left(\tilde{Q}_\pi\left(\left[x, y^{<t}\right], y^t\right) - \Phi^\mu\left(\tilde{V}_\pi\left(\left[x, y^{<t}\right]\right)\right)\right) - \gamma^T\,\Phi^\mu\left(\tilde{V}_\pi\left(\left[x, y^{<T+1}\right]\right)\right).
\end{aligned} \tag{27}$$

Next, note that $y^T = \text{EOS}$ denotes the end of the text sequence. Therefore,

$$V_\pi\left(\left[x, y^{<T+1}\right]\right) = \mathbb{E}_\pi\left[\sum_{k=0}^{\infty} \gamma^k R\left(\left[x, y^{<T+1+k}\right], y^{T+1+k}\right) \mid s_t = \left[x, y^{<T+1}\right]\right] = 0. \qquad (28)$$

Furthermore, we have

$$r = \Phi^\mu\left(\tilde{V}_\pi\left([x]\right)\right) + \sum_{t=1}^{T} \gamma^{t-1}\left(\tilde{Q}_\pi\left(\left[x, y^{<t}\right], y^t\right) - \Phi^\mu\left(\tilde{V}_\pi\left(\left[x, y^{<t}\right]\right)\right)\right). \qquad (29)$$

So, we definite the risk-aware advantage function as $\tilde{A}_\pi\left(\left[x, y^{<t}\right], z\right) = \tilde{Q}_\pi\left(\left[x, y^{<t}\right], z\right) - \Phi^\mu\left(\tilde{V}_\pi\left(\left[x, y^{<t}\right]\right)\right)$, where $z \sim \pi_\theta\left(\cdot \mid \left[x, y^{<t}\right]\right)$.

### B.3    The Proof of Lemma 3.3

**Lemma 3.3 Restated.**    Given two policies $\pi$ and $\pi'$, if for any state $s_t = \left[x, y^{<t}\right], \mathbb{E}_{z \sim \pi'}\left[\tilde{A}_\pi\left(\left[x, y^{<t}\right], z\right)\right] \geq 0$ holds, then we can conclude:

$$\mathbb{E}_{x \sim \mathcal{D}}\left[\tilde{V}_{\pi'}\left([x]\right)\right] \geq \mathbb{E}_{x \sim \mathcal{D}}\left[\tilde{V}_\pi\left([x]\right)\right].$$

*Proof.* Let $\tau := \left(x, y^1, y^2, \ldots, y^T\right)$ denote a trajectory, where the expectation $\mathbb{E}_{\tau \mid \pi'}[\cdot]$ is taken over trajectories generated by policy $\pi'$. We then have

$$\begin{aligned}
&\mathbb{E}_{x \sim \mathcal{D}}\left[\tilde{V}_{\pi'}\left([x]\right)\right] - \mathbb{E}_{x \sim \mathcal{D}}\left[\tilde{V}_\pi\left([x]\right)\right] \\
=&\mathbb{E}_{\tau \mid \pi'}\left[\sum_{t=1}^{T} \gamma^{t-1}\left(R\left(\left[x, y^{<t}\right], y^t\right) + \gamma\,\Phi^\mu\left(\tilde{V}_\pi\left(\left[x, y^{<t+1}\right]\right)\right)\right) - \tilde{V}_\pi\left([x]\right)\right] \\
=&\mathbb{E}_{\tau \mid \pi'}\left[\sum_{t=1}^{T} \gamma^{t-1}\left(R\left(\left[x, y^{<t}\right], y^t\right) + \gamma\,\Phi^\mu\left(\tilde{V}_\pi\left(\left[x, y^{<t+1}\right]\right)\right) - \Phi^\mu\left(\tilde{V}_\pi\left(\left[x, y^{<t}\right]\right)\right)\right)\right] \\
=&\mathbb{E}_{\tau \mid \pi'}\left[\sum_{t=1}^{T} \gamma^{t-1}\left(\tilde{A}_\pi\left(\left[x, y^{<t}\right], y^t\right)\right)\right] \\
=&\mathbb{E}_{\tau \mid \pi'}\left[\sum_{t=1}^{T} \gamma^{t-1}\left(\mathbb{E}_{y^t \sim \pi'}\left[\tilde{A}_\pi\left(\left[x, y^{<t}\right], y^t\right)\right]\right)\right].
\end{aligned} \qquad (30)$$

Since for any state $s_t = \left[x, y^{<t}\right], \mathbb{E}_{z \sim \pi'}\left[\tilde{A}_\pi\left(\left[x, y^{<t}\right], z\right)\right] \geq 0$, so we can obtain

$$\mathbb{E}_{x \sim \mathcal{D}}\left[\tilde{V}_{\pi'}\left([x]\right)\right] - \mathbb{E}_{x \sim \mathcal{D}}\left[\tilde{V}_\pi\left([x]\right)\right] \geq 0.$$

This completes the proof of Lemma 3.3. $\qquad\qquad\qquad\qquad\qquad\qquad\qquad\qquad\square$

### B.4    The Proof of Lemma 3.4

**Lemma 3.4 Restated.** The constrained problem in Equation (8) has the closed-form solution:

$$\pi_\theta^*\left(z \mid \left[x, y^{<t}\right]\right) = \frac{\pi_{\text{ref}}\left(z \mid \left[x, y^{<t}\right]\right)\exp\left(\frac{1}{\beta}\tilde{Q}_{\pi_{\text{ref}}}\left(\left[x, y^{<t}\right], z\right)\right)}{Z\left(\left[x, y^{<t}\right]; \beta\right)},$$

where $Z\left(\left[x, y^{<t}\right]; \beta\right) = \mathbb{E}_{z \sim \pi_{\text{ref}}\left(\cdot \mid \left[x, y^{<t}\right]\right)}e^{\frac{1}{\beta}\tilde{Q}_{\pi_{\text{ref}}}\left(\left[x, y^{<t}\right], z\right)}$ is the partition function.

*Proof.*

$$\max_{\pi_\theta} \mathbb{E}_{z \sim \pi_\theta(\cdot|[x,y^{<t}])} \tilde{A}_{\pi_{\mathrm{ref}}}\left(\left[x, y^{<t}\right], z\right) - \beta D_{\mathrm{KL}}\left(\pi_\theta\left(\cdot \mid \left[x, y^{<t}\right]\right) \| \pi_{\mathrm{ref}}\left(\cdot \mid \left[x, y^{<t}\right]\right)\right)$$

$$= \max_{\pi_\theta} \mathbb{E}_{z \sim \pi_\theta(\cdot|[x,y^{<t}])}\left(\left(\tilde{Q}_{\pi_{\mathrm{ref}}}\left(\left[x, y^{<t}\right], z\right) - \tilde{V}_{\pi_{\mathrm{ref}}}\left(\left[x, y^{<t}\right]\right)\right) + \beta \log\left(\frac{\pi_{\mathrm{ref}}\left(z \mid \left[x, y^{<t}\right]\right)}{\pi_\theta\left(z \mid \left[x, y^{<t}\right]\right)}\right)\right)$$

$$= \max_{\pi_\theta} \beta \mathbb{E}_{z \sim \pi_\theta(\cdot|[x,y^{<t}])} \log\left(\frac{\pi_{\mathrm{ref}}\left(z \mid \left[x, y^{<t}\right]\right) e^{\frac{1}{\beta}\tilde{Q}_{\pi_{\mathrm{ref}}}\left(\left[x,y^{<t}\right],z\right)}}{\pi_\theta\left(z \mid \left[x, y^{<t}\right]\right)}\right) - \tilde{V}_{\pi_{\mathrm{ref}}}\left(\left[x, y^{<t}\right]\right)$$

$$= \max_{\pi_\theta} \beta \mathbb{E}_{z \sim \pi_\theta(\cdot|[x,y^{<t}])} \log\left(\frac{\pi_{\mathrm{ref}}\left(z \mid \left[x, y^{<t}\right]\right) e^{\frac{1}{\beta}\tilde{Q}_{\pi_{\mathrm{ref}}}\left(\left[x,y^{<t}\right],z\right)}}{Z\left(\left[x, y^{<t}\right]; \beta\right) \pi_\theta\left(z \mid \left[x, y^{<t}\right]\right)}\right)$$

$$- \tilde{V}_{\pi_{\mathrm{ref}}}\left(\left[x, y^{<t}\right]\right) + \beta \log Z\left(\left[x, y^{<t}\right]; \beta\right)$$

$$= \max_{\pi_\theta} -\beta D_{\mathrm{KL}}\left(\pi_\theta\left(z \mid \left[x, y^{<t}\right]\right) \| \frac{\pi_{\mathrm{ref}}\left(z \mid \left[x, y^{<t}\right]\right) e^{\frac{1}{\beta}\tilde{Q}_{\pi_{\mathrm{ref}}}\left(\left[x,y^{<t}\right],z\right)}}{Z\left(\left[x, y^{<t}\right]; \beta\right)}\right)$$

$$- \tilde{V}_{\pi_{\mathrm{ref}}}\left(\left[x, y^{<t}\right]\right) + \beta \log Z\left(\left[x, y^{<t}\right]; \beta\right),$$

$$\tag{31}$$

where $Z\left(\left[x, y^{<t}\right]; \beta\right)$ is the partition function:

$$Z\left(\left[x, y^{<t}\right]; \beta\right) = \mathbb{E}_{z \sim \pi_{\mathrm{ref}}(\cdot|[x,y^{<t}])} \exp\left(\frac{1}{\beta}\tilde{Q}_{\pi_{\mathrm{ref}}}\left(\left[x, y^{<t}\right], z\right)\right). \tag{32}$$

Then, we can derive the relationship between the optimal policy and the state-action function:

$$\pi_\theta^*\left(z \mid \left[x, y^{<t}\right]\right) = \frac{\pi_{\mathrm{ref}}\left(z \mid \left[x, y^{<t}\right]\right) \exp\left(\frac{1}{\beta}\tilde{Q}_{\pi_{\mathrm{ref}}}\left(\left[x, y^{<t}\right], z\right)\right)}{Z\left(\left[x, y^{<t}\right]; \beta\right)}.$$

This completes the proof of Lemma 3.4. $\qquad\square$

### B.5 The Proof of Lemma 3.5

**Lemma 3.5 Restated.** Given a reward function $r(x, y)$ over the entire prompt-response, based on the relationship between token-wise rewards and the reward function $r(x, y) = \sum_{t=1}^{T} \gamma^{t-1} R\left(\left[x, y^{<t}\right], y^t\right)$, we can establish the equivalence between the Bradley-Terry model and the Regret Preference Model, i.e.,

$$P_{\mathrm{BT}}\left(y_1 \succ y_2 \mid x\right) = \sigma\left(\sum_{t=1}^{T_1} \gamma^{t-1} \tilde{A}_\pi\left(\left[x, y_1^{<t}\right], y_1^t\right) - \sum_{t=1}^{T_2} \gamma^{t-1} \tilde{A}_\pi\left(\left[x, y_2^{<t}\right], y_2^t\right)\right),$$

where $\sigma(z) = 1/\left(1 + \exp(-z)\right)$ is the logistic sigmoid function for any random variable $z$.

*Proof.* Recalling to the BT model in Equation (1)

$$P_{\mathrm{BT}}\left(y_1 \succ y_2 \mid x\right) = \frac{\exp\left(r\left(x, y_1\right)\right)}{\exp\left(r\left(x, y_1\right)\right) + \exp\left(r\left(x, y_2\right)\right)}, \tag{33}$$

and the equivalence between prompt-response reward and the risk-aware advantage function:

$$r = \Phi^\mu\left(\tilde{V}_\pi\left(\left[x\right]\right)\right) + \sum_{t=1}^{T} \gamma^{t-1}\left(\tilde{Q}_\pi\left(\left[x, y^{<t}\right], y^t\right) - \Phi^\mu\left(\tilde{V}_\pi\left(\left[x, y^{<t}\right]\right)\right)\right)$$

$$= \Phi^\mu\left(\tilde{V}_\pi\left(\left[x\right]\right)\right) + \sum_{t=1}^{T} \gamma^{t-1} \tilde{A}_\pi\left(\left[x, y^{<t}\right], y^t\right).$$

Then, we have

$$P_{\mathrm{BT}}\left(y_1 \succ y_2 \mid x\right) = \sigma\left(\sum_{t=1}^{T_1} \gamma^{t-1} \tilde{A}_\pi\left(\left[x, y_1^{<t}\right], y_1^t\right) - \sum_{t=1}^{T_2} \gamma^{t-1} \tilde{A}_\pi\left(\left[x, y_2^{<t}\right], y_2^t\right)\right).$$

This completes the proof of Lemma 3.5. $\qquad\square$

## B.6 The Proof of Theorem 3.6

**Theorem 3.6 Restated.** Given prompts $x$ and pairwise responses $(y_1, y_2)$, and the risk-aware objective function in Equation (8), the Bradley-Terry model expresses the human preference probability in terms of the risk-aware optimal policy $\pi_\theta^*$ and reference policy $\pi_{\text{ref}}$:

$$P_{\text{BT}}^* (y_1 \succ y_2 \mid x) = \sigma \left( u^* (x, y_1, y_2) - \delta^* (x, y_1, y_2) \right),$$

where $u(x, y_1, y_2)$ represents the difference in implicit rewards defined by the risk-aware policy $\pi_\theta^*$ and the reference policy $\pi_{\text{ref}}$, weighted by $\beta$, represented as

$$u(x, y_1, y_2) = \beta \log \frac{\pi_\theta(y_1 \mid x)}{\pi_{\text{ref}}(y_1 \mid x)} - \beta \log \frac{\pi_\theta(y_2 \mid x)}{\pi_{\text{ref}}(y_2 \mid x)},$$

and $\delta(x, y_1, y_2)$ represents the difference in sequential risk ratio between two pairs $(x, y_1)$ and $(x, y_2)$, expressed as

$$\delta(x, y_1, y_2) = \beta D_{\text{SeqRR}}(x, y_2; \pi_{\text{ref}} \mid \pi_\theta) - \beta D_{\text{SeqRR}}(x, y_1; \pi_{\text{ref}} \mid \pi_\theta).$$

*Proof.* According to the Lemma 3.4, we have

$$\pi_\theta^* \left( z \mid [x, y^{<t}] \right) = \frac{\pi_{\text{ref}} \left( z \mid [x, y^{<t}] \right) \exp \left( \frac{1}{\beta} \tilde{Q}_{\pi_{\text{ref}}} \left( [x, y^{<t}], z \right) \right)}{Z \left( [x, y^{<t}]; \beta \right)}, \tag{34}$$

where $Z \left( [x, y^{<t}]; \beta \right) = \mathbb{E}_{z \sim \pi_{\text{ref}}(\cdot \mid [x, y^{<t}])} e^{\frac{1}{\beta} \tilde{Q}_{\pi_{\text{ref}}} \left( [x, y^{<t}], z \right)}$ is the partition function. Rearrange Equation (34), we obtain

$$\tilde{Q}_{\pi_{\text{ref}}} \left( [x, y^{<t}], z \right) = \beta \log \frac{\pi_\theta^* \left( z \mid [x, y^{<t}] \right)}{\pi_{\text{ref}} \left( z \mid [x, y^{<t}] \right)} + \beta \log Z \left( [x, y^{<t}]; \beta \right). \tag{35}$$

From Lemma 3.5, we can get

$$P_{\text{BT}}(y_1 \succ y_2 \mid x) = \sigma \left( \sum_{t=1}^{T_1} \left( \gamma^{t-1} \tilde{A}_\pi \left( [x, y_1^{<t}], y_1^t \right) \right) - \sum_{t=1}^{T_2} \left( \gamma^{t-1} \tilde{A}_\pi \left( [x, y_2^{<t}], y_2^t \right) \right) \right). \tag{36}$$

By leveraging Equation (35), we can derive

$$\sum_{t=1}^{T} \gamma^{t-1} \tilde{A}_{\pi_{\text{ref}}} \left( [x, y^{<t}], y^t \right)$$

$$= \sum_{t=1}^{T} \gamma^{t-1} \left( Q_{\pi_{\text{ref}}} \left( [x, y^{<t}], y^t \right) - \Phi^\mu \left( \tilde{V}_{\pi_{\text{ref}}} \left( [x, y^{<t}] \right) \right) \right)$$

$$= \sum_{t=1}^{T} \gamma^{t-1} \left( \tilde{Q}_{\pi_{\text{ref}}} \left( [x, y^{<t}], y^t \right) - \Phi^\mu \left( \tilde{Q}_{\pi_{\text{ref}}} \left( [x, y^{<t}], z \right) \right) \right) \tag{37}$$

$$= \sum_{t=1}^{T} \gamma^{t-1} \left( \beta \log \frac{\pi_\theta^* \left( y^t \mid [x, y^{<t}] \right)}{\pi_{\text{ref}} \left( y^t \mid [x, y^{<t}] \right)} + \beta \log Z \left( [x, y^{<t}]; \beta \right) \right.$$

$$\left. - \Phi^\mu \left( \beta \log \frac{\pi_\theta^* \left( z \mid [x, y^{<t}] \right)}{\pi_{\text{ref}} \left( z \mid [x, y^{<t}] \right)} + \beta \log Z \left( [x, y^{<t}]; \beta \right) \right) \right).$$

Note that

$$\mathbb{E}_{z \sim \pi_{\text{ref}}} \left[ \beta \log Z \left( [x, y^{<t}]; \beta \right) \right] = \beta \log Z \left( [x, y^{<t}]; \beta \right).$$

Therefore,

$$\sum_{t=1}^{T} \gamma^{t-1} \tilde{A}_{\pi_{\text{ref}}} \left( [x, y^{<t}], y^t \right)$$

$$= \beta \sum_{t=1}^{T} \gamma^{t-1} \left( \log \frac{\pi_\theta^* \left( y^t \mid [x, y^{<t}] \right)}{\pi_{\text{ref}} \left( y^t \mid [x, y^{<t}] \right)} - \Phi_{z \sim \pi_{\text{ref}}}^\mu \left( \log \frac{\pi_\theta^* \left( z \mid [x, y^{<t}] \right)}{\pi_{\text{ref}} \left( z \mid [x, y^{<t}] \right)} \right) \right) \tag{38}$$

$$= \beta \sum_{t=1}^{T} \gamma^{t-1} \log \frac{\pi_\theta^* \left( y^t \mid [x, y^{<t}] \right)}{\pi_{\text{ref}} \left( y^t \mid [x, y^{<t}] \right)} + \beta \sum_{t=1}^{T} \gamma^{t-1} \Phi_{z \sim \pi_{\text{ref}}}^\mu \left( \log \frac{\pi_{\text{ref}} \left( z \mid [x, y^{<t}] \right)}{\pi_\theta^* \left( z \mid [x, y^{<t}] \right)} \right).$$

When substituting $\gamma = 1$ into the expression, we obtain a more concise form:

$$
\begin{aligned}
\sum_{t=1}^{T} \tilde{A}_{\pi_{\text{ref}}} \left( \left[ x, y^{<t} \right], y^t \right) =& \beta \sum_{t=1}^{T} \log \frac{\pi_\theta^* \left( y^t \mid [x, y^{<t}] \right)}{\pi_{\text{ref}} \left( y^t \mid [x, y^{<t}] \right)} + \beta \sum_{t=1}^{T} \Phi_{z \sim \pi_{\text{ref}}}^{\mu} \left( \log \frac{\pi_{\text{ref}} \left( z \mid [x, y^{<t}] \right)}{\pi_\theta^* \left( z \mid [x, y^{<t}] \right)} \right) \\
=& \beta \left( \log \frac{\pi_\theta^* \left( y \mid x \right)}{\pi_{\text{ref}} \left( y \mid x \right)} + D_{\text{SeqRR}} \left( x, y; \pi_{\text{ref}} \mid \pi_\theta^* \right) \right),
\end{aligned}
\tag{39}
$$

where $D_{\text{SeqRR}} \left( x, y; \pi_{\text{ref}} \mid \pi_\theta \right) = \sum_{t=1}^{T} \Phi_{z \sim \pi_{\text{ref}}}^{\mu} \left( \log \frac{\pi_{\text{ref}} \left( z \mid [x, y^{<t}] \right)}{\pi_\theta \left( z \mid [x, y^{<t}] \right)} \right)$.

Then, we let

$$
u \left( x, y_1, y_2 \right) = \beta \log \frac{\pi_\theta \left( y_1 \mid x \right)}{\pi_{\text{ref}} \left( y_1 \mid x \right)} - \beta \log \frac{\pi_\theta \left( y_2 \mid x \right)}{\pi_{\text{ref}} \left( y_2 \mid x \right)},
\tag{40}
$$

$$
\delta \left( x, y_1, y_2 \right) = \beta D_{\text{SeqRR}} \left( x, y_2; \pi_{\text{ref}} \mid \pi_\theta \right) - \beta D_{\text{SeqRR}} \left( x, y_1; \pi_{\text{ref}} \mid \pi_\theta \right).
\tag{41}
$$

Substituting Equation (39) into Equation (36), we arrive at

$$
P_{\text{BT}}^* \left( y_1 \succ y_2 \mid x \right) = \sigma \left( u^* \left( x, y_1, y_2 \right) - \delta^* \left( x, y_1, y_2 \right) \right).
$$

This completes the proof Theorem 3.6. $\qquad\square$

### B.7 Algorithm

In this subsection, we provide the main pseudocode for Risk-aware Direct Preference Optimization (Ra-DPO), as outlined in Algorithm 1.

---
**Algorithm 1** Risk-aware Direct Preference Optimization (Ra-DPO)
---
**Input:** Reference model $\pi_{\text{ref}}$, Policy model $\pi_\theta$, Coefficient $\alpha$, $\beta$, Risk control parameter $\mu$, Learning rate $\eta$
**Input:** Dataset $\mathcal{D} = \left\{ (x, y_w, y_l)^i \right\}_{i=1}^{N}$ of size $N$, Method $\mathcal{M}$
**Initialize:** $\pi_\theta \leftarrow \pi_{\text{ref}}$
**for** each epoch **do**
    Sample mini-batch $\mathcal{D}_m = \{(x, y_w, y_l)^m\}_{m=1}^{M}$ from $\mathcal{D}$
    Predict the probabilities $\pi_\theta (y_w \mid x)$ and $\pi_\theta (y_l \mid x)$ for $(x, y_w, y_l)$ in the mini-batch $\mathcal{D}_m$ using the policy model
    Predict the probabilities $\pi_{\text{ref}} (y_w \mid x)$ and $\pi_{\text{ref}} (y_l \mid x)$ for $(x, y_w, y_l)$ in the mini-batch $\mathcal{D}_m$ using the reference model
    Calculate the function $u (x, y_w, y_l) = \beta \log \frac{\pi_\theta (y_w \mid x)}{\pi_{\text{ref}} (y_w \mid x)} - \beta \log \frac{\pi_\theta (y_l \mid x)}{\pi_{\text{ref}} (y_l \mid x)}$
    Compute the sequential risk ratio $D_{\text{SeqRR}} (x, y_w; \pi_{\text{ref}} \mid \pi_\theta)$ for $(x, y_w)$ in the mini-batch $\mathcal{D}_m$
    Compute the sequential risk ratio $D_{\text{SeqRR}} (x, y_l; \pi_{\text{ref}} \mid \pi_\theta)$ for $(x, y_l)$ in the mini-batch $\mathcal{D}_m$
    **if** Method $\mathcal{M}$ is Ra-DPO$_1$ **then**
        Calculate $\delta (x, y_w, y_l) = \beta D_{\text{SeqRR}} (x, y_l; \pi_{\text{ref}} \mid \pi_\theta) - \beta D_{\text{SeqRR}} (x, y_w; \pi_{\text{ref}} \mid \pi_\theta)$
        $\theta \leftarrow \theta + \eta \nabla_\theta \mathbb{E}_{(x, y_w, y_l) \sim \mathcal{D}_m} \left[ \log \sigma \left( u (x, y_w, y_l) - \delta (x, y_w, y_l) \right) \right]$
    **else** {Method $\mathcal{M}$ is Ra-DPO$_2$}
        Calculate $\delta_2 (x, y_w, y_l) = \beta D_{\text{SeqRR}} (x, y_l; \pi_{\text{ref}} \mid \pi_\theta) - \text{sg} \left( \beta D_{\text{SeqRR}} (x, y_w; \pi_{\text{ref}} \mid \pi_\theta) \right)$
        $\theta \leftarrow \theta + \eta \nabla_\theta \mathbb{E}_{(x, y_w, y_l) \sim \mathcal{D}_m} \left[ \log \sigma \left( u (x, y_w, y_l) - \alpha \delta_2 (x, y_w, y_l) \right) \right]$
    **end if**
**end for**

---

## C Supplementary Materials for Section 4

### C.1 Experiments compute resources

All reported results of our algorithm and baseline algorithms are trained using $4 \times$ A100 GPUs, each with 40GB of memory.

## C.2 Assets

We have compiled the datasets, models, and benchmark codes used in this paper and express our gratitude to all relevant sources.

**Dataset:**

- IMDb Dataset [32]: `https://huggingface.co/datasets/stanfordnlp/imdb`

- Anthropic HH Dataset [33]: `https://huggingface.co/datasets/Anthropic/hh-rlhf`

- AlpacaEval [34]: `https://huggingface.co/datasets/tatsu-lab/alpaca_eval`

**Model:**

- GPT-2 Large [36]: `https://huggingface.co/openai-community/gpt2-large`

- Gpt2-large-imdb-fine-tuned: `https://huggingface.co/insub/gpt2-large-IMDb-fine-tuned`

- Pythia-1.4B [37]: `https://huggingface.co/EleutherAI/pythia-1.4b`

- Pythia-2.8B [37]: `https://huggingface.co/EleutherAI/pythia-2.8b`

- Oasst-sft-4-pythia-12b-epoch-3.5: `https://huggingface.co/OpenAssistant/oasst-sft-4-pythia-12b-epoch-3.5`

**Code:**

- We trained Ra-DPO and the baseline models based on the original KTO implementation `https://github.com/ContextualAI/HALOs`, and our code can be found in the supplemental material.

## C.3 Experimental Details

In our experiments, we followed the original KTO implementation for the main parameter settings, and both Ra-DPO and the baseline models used the same hyperparameters, as detailed in Table 2-3.

Table 2: Hyperparameters in loss functions for different algorithms.

| Method | $\beta$ | $\alpha$ | $\mu$ |
|---|---|---|---|
| DPO | 0.1 | - | - |
| PPO | - | - | - |
| KTO | 0.1 | - | - |
| TDPO$_1$ | 0.1 | - | - |
| TDPO$_2$ | 0.1 | $\{0.3, 0.5, 0.7, 0.9\}$ | - |
| Ra-DPO$_1$ | 0.1 | - | - |
| Ra-DPO$_2$ | 0.1 | $\{0.3, 0.5, 0.7, 0.9\}$ | CVaR: $\{0.99, 0.98, 0.97, 0.95\}$
ERM: $\{9, 7, 5\}$ |

Table 3: Hyperparameters in network training.

| Parameter | value |
|---|---|
| max length | 512 |
| max prompt length | 256 |
| gradient accumulation steps | 4 |
| learning rate | $5 \times 10^{-6}$ |
| optimizer | AdamW |

## C.4 Additional Experimental Results

Here, we provide some additional experimental results, which are illustrated in Figures 5-14.

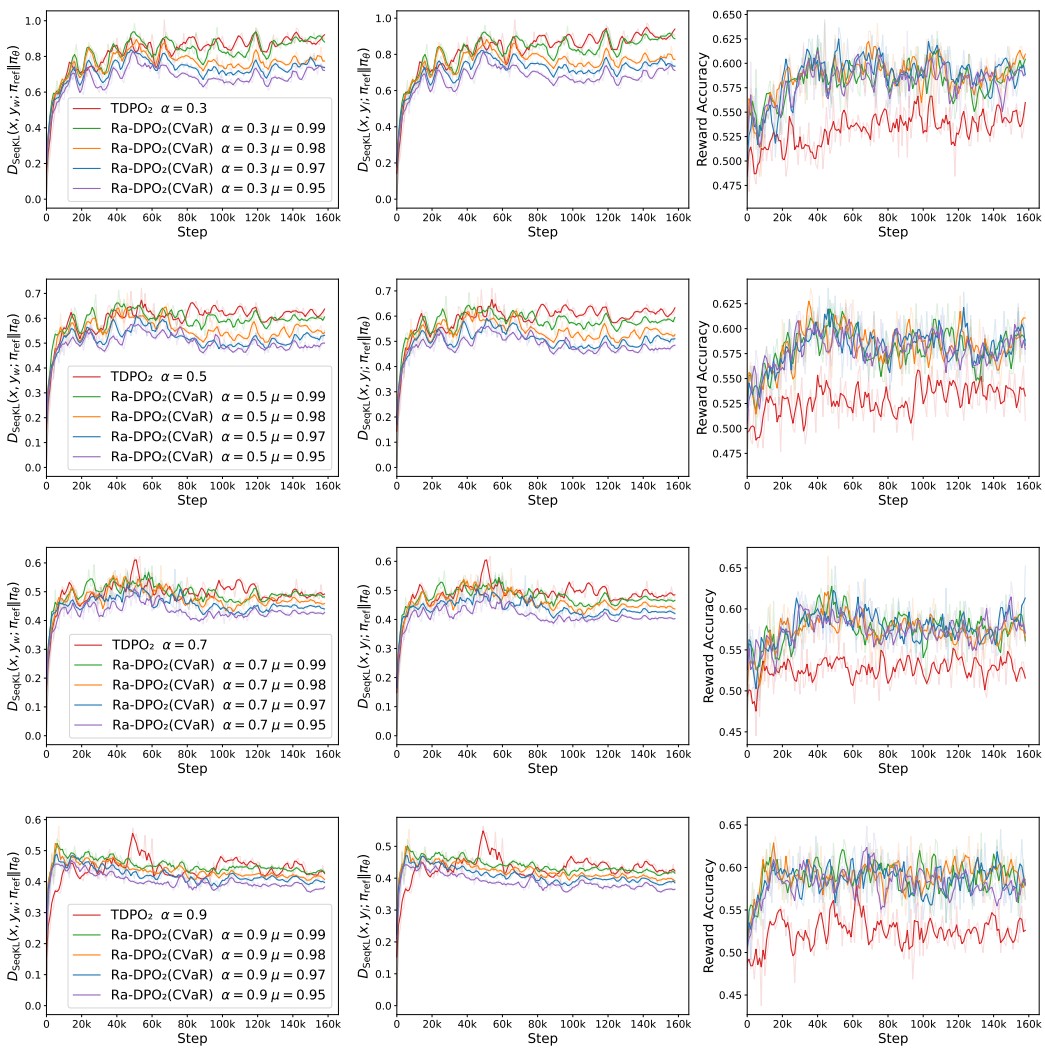

Figure 5: The experiment on the Anthropic HH dataset with Pythia-1.4B serving as the base model. **Left** and **Middle** present the progression of sequential KL divergence (the lower the better) for both preferred and dispreferred responses. **Right** illustrates reward accuracy curves (the higher the better).

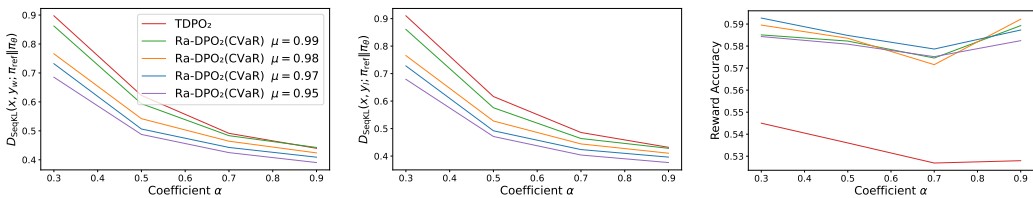

Figure 6: The experiment on the Anthropic HH dataset with Pythia-1.4B serving as the base model. **Left** and **Middle** presents the sequential KL divergence (the lower the better) for preferred and dispreferred responses, while **Right** presents the reward accuracy curves (the higher the better) under $\alpha = \{0.3, 0.5, 0.7, 0.9\}$.

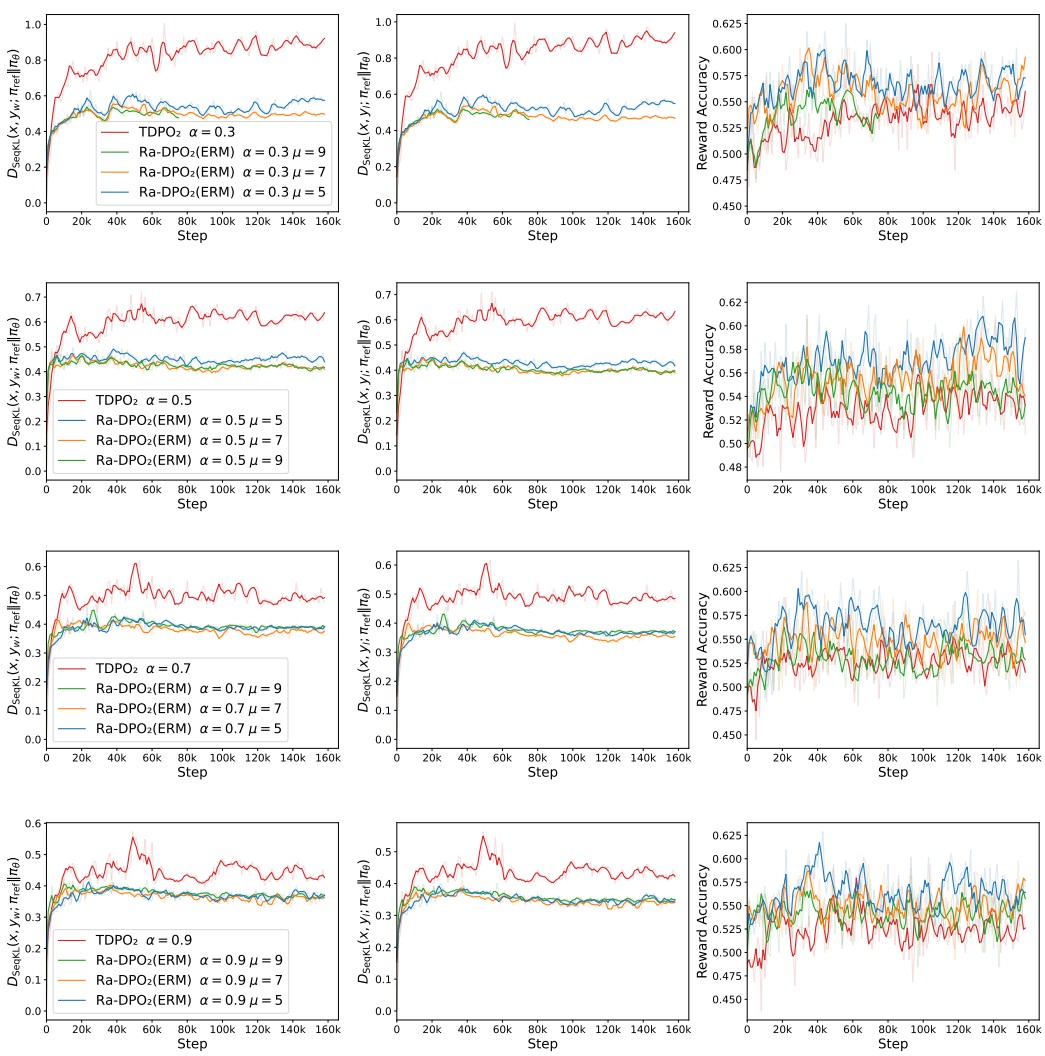

Figure 7: The experiment on the Anthropic HH dataset with Pythia-1.4B serving as the base model. **Left** and **Middle** present the progression of sequential KL divergence (the lower the better) for both preferred and dispreferred responses. **Right** illustrates reward accuracy curves (the higher the better).

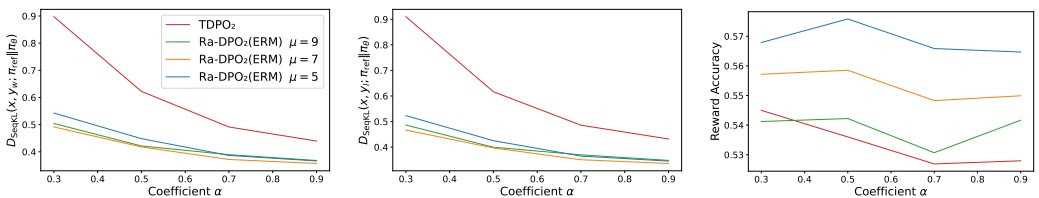

Figure 8: The experiment on the Anthropic HH dataset with Pythia-1.4B serving as the base model. **Left** and **Middle** presents the sequential KL divergence (the lower the better) for preferred and dispreferred responses, while **Right** presents the reward accuracy curves (the higher the better) under $\alpha = \{0.3, 0.5, 0.7, 0.9\}$.

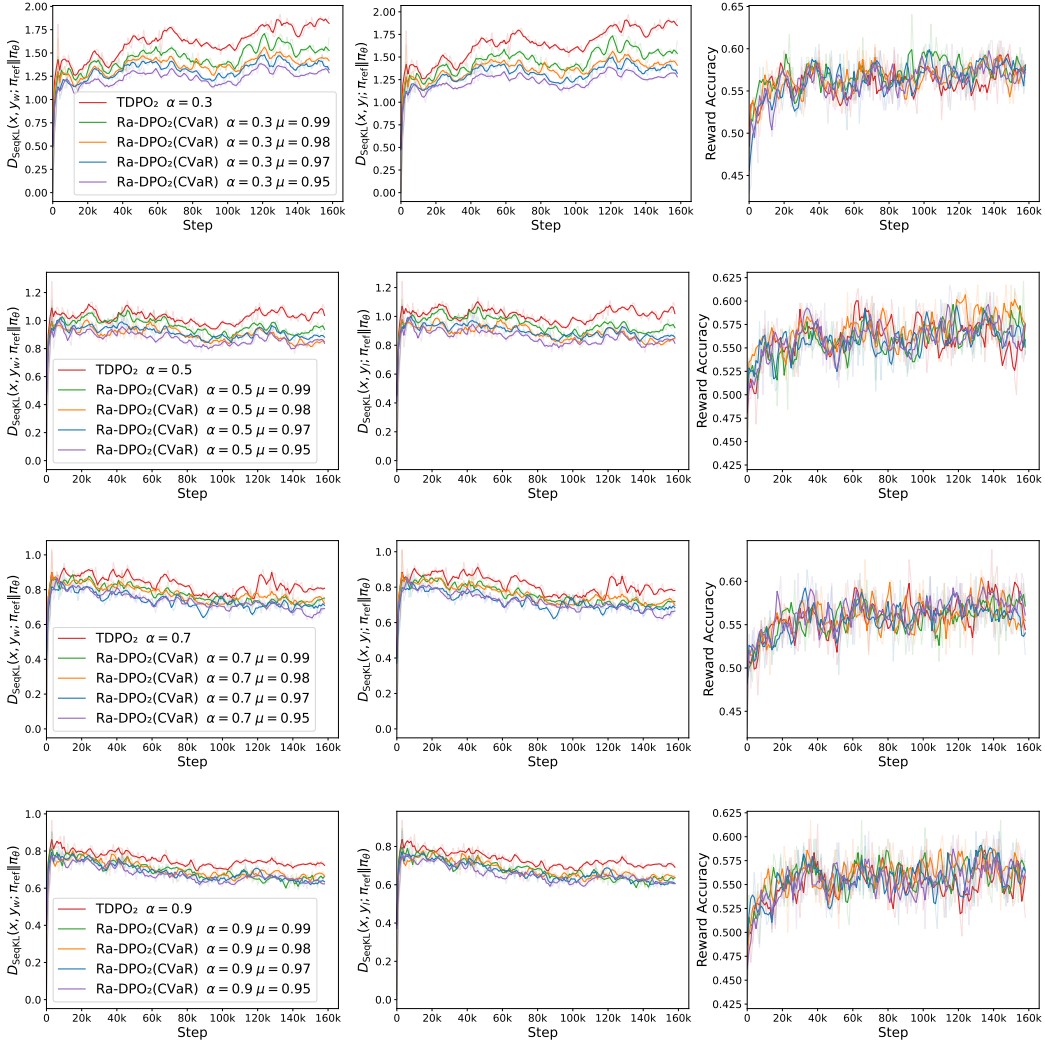

Figure 9: The experiment on the Anthropic HH dataset with Pythia-2.8B serving as the base model. **Left** and **Middle** present the progression of sequential KL divergence (the lower the better) for both preferred and dispreferred responses. **Right** illustrates reward accuracy curves (the higher the better).

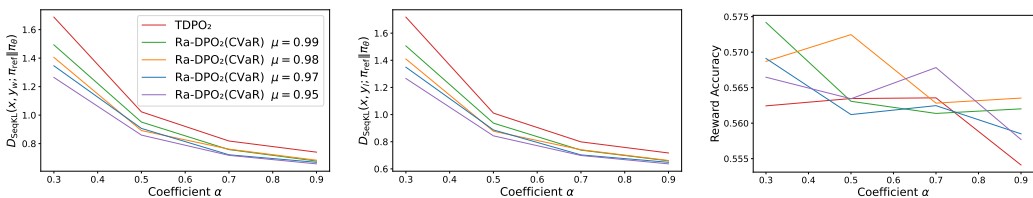

Figure 10: The experiment on the Anthropic HH dataset with Pythia-2.8B serving as the base model. **Left** and **Middle** presents the sequential KL divergence (the lower the better) for preferred and dispreferred responses, while **Right** presents the reward accuracy curves (the higher the better) under $\alpha = \{0.3, 0.5, 0.7, 0.9\}$.

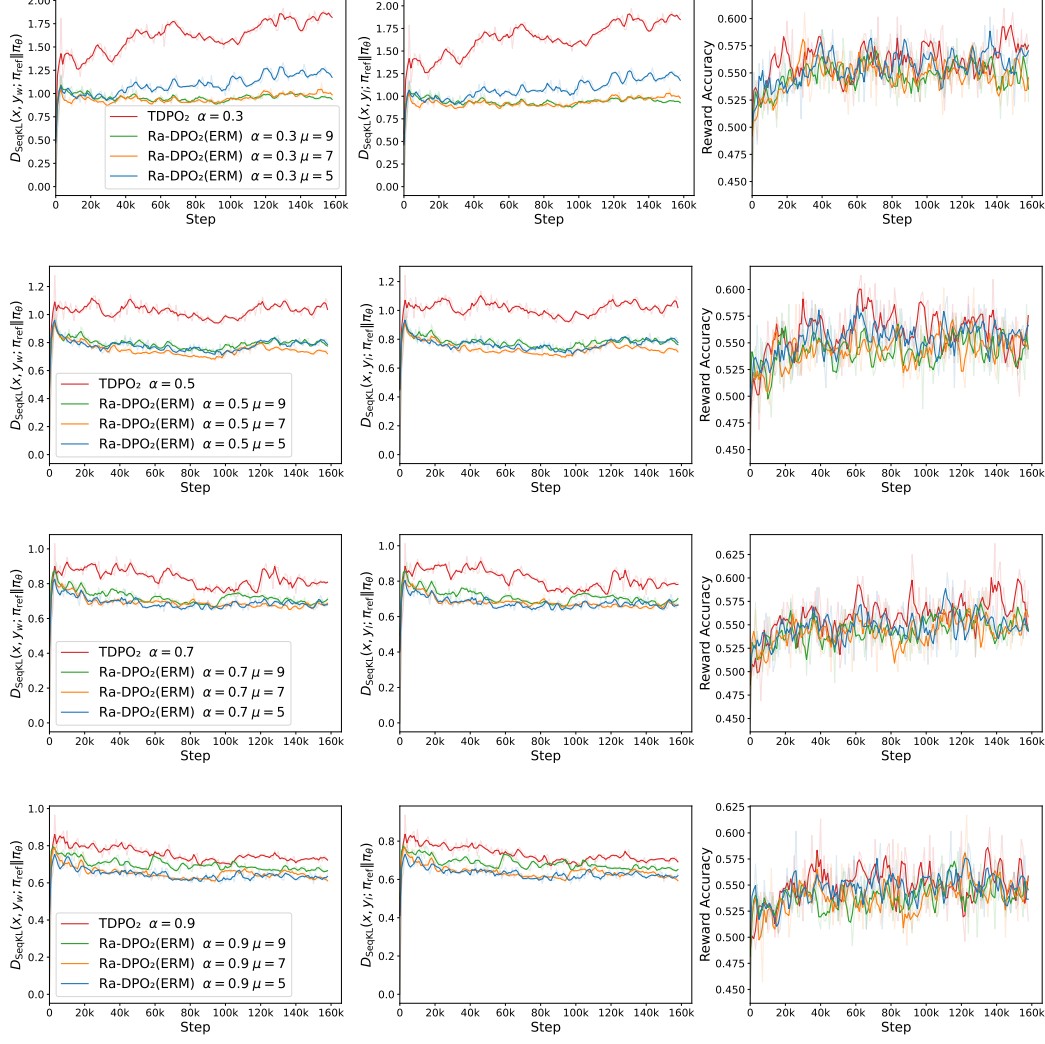

Figure 11: The experiment on the Anthropic HH dataset with Pythia-2.8B serving as the base model. **Left** and **Middle** present the progression of sequential KL divergence (the lower the better) for both preferred and dispreferred responses. **Right** illustrates reward accuracy curves (the higher the better).

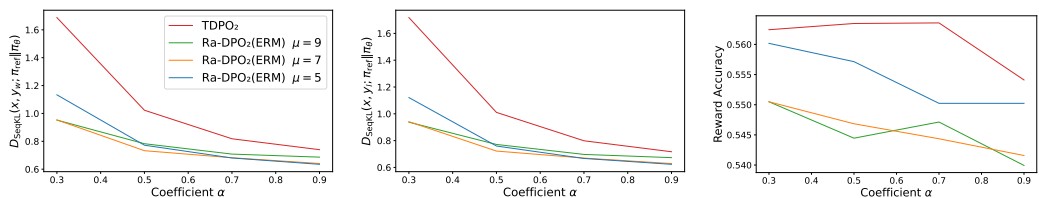

Figure 12: The experiment on the Anthropic HH dataset with Pythia-2.8B serving as the base model. **Left** and **Middle** presents the sequential KL divergence (the lower the better) for preferred and dispreferred responses, while **Right** presents the reward accuracy curves (the higher the better) under $\alpha = \{0.3, 0.5, 0.7, 0.9\}$.

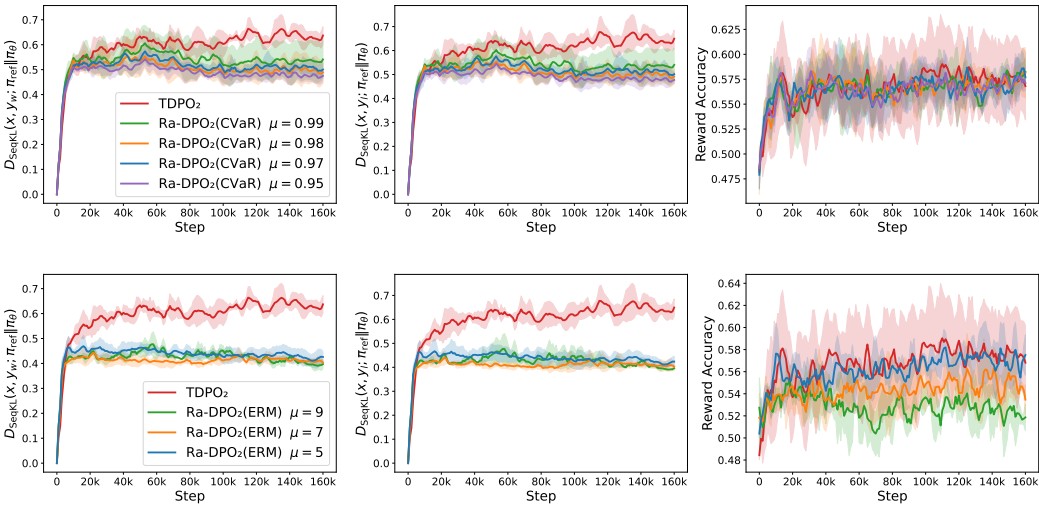

Figure 13: The experiment on the Anthropic HH dataset with Pythia-1.4B serving as the base model. **Left** and **Middle** present the progression of sequential KL divergence (the lower the better) for both preferred and dispreferred responses. **Right** illustrates reward accuracy curves (the higher the better). For all algorithms, we report the average performance (solid line) across three random seeds, with the shaded region representing one standard deviation around the mean.

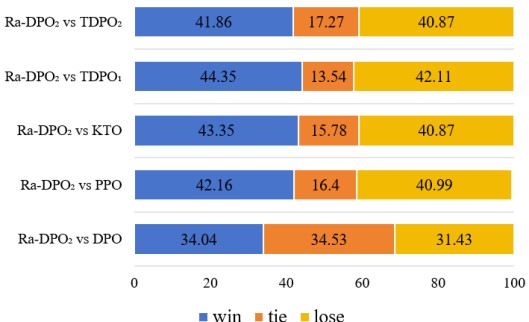

Figure 14: AlpacaEval comparison between DPO, PPO, TDPO$_1$, TDPO$_2$, and Ra-DPO$_2$ methods. The win, tie, and lose rates are evaluated based on *oasst-pythia-12b*.

- Figures 5-8 illustrate the experiment on the Anthropic HH dataset with Pythia-1.4B serving as the base model. We implemented TDPO$_2$, and different versions of Ra-DPO$_2$ with respect to the parameters $\alpha$ and $\mu$.

- Figures 9-12 show corresponding results using Pythia-2.8B as the base model. The same set of algorithms was evaluated under varying $\alpha$ and $\mu$ configurations.

- Figure 13 illustrates the experiment on the Anthropic HH dataset with Pythia-1.4B serving as the base model. Let $\alpha = 0.5$, we implemented TDPO$_2$, and different versions of Ra-DPO$_2$ with respect to the risk control parameter $\mu$. In the figure, for all algorithms, we report the average performance (solid line) across three random seeds, with the shaded region representing one standard deviation around the mean. We aim to highlight the statistically significant improvements achieved by the proposed method, although training large-scale models entails substantial computational costs in terms of time and resources. The figure illustrates that, under both the CVaR-based nested risk measure and the ERM-based risk measure with $\mu = 5$, the proposed algorithm achieves reward accuracy comparable to

that of the baseline method but with greater stability, while maintaining a consistently low sequential KL divergence.

- Figure 14 illustrates the comparison between DPO, PPO, $TDPO_1$, $TDPO_2$, and $Ra\text{-}DPO_2$ methods through AlpacaEval. It presents a straightforward result: Compared to the baseline algorithms, $Ra\text{-}DPO_2$ achieves a high winrate, demonstrating superior performance in assisting LLMs to generate high-quality responses.

