# OpenReview forum: "Risk-aware Direct Preference Optimization under Nested Risk Measure"
_NeurIPS.cc/2025/Conference — NeurIPS 2025 poster_

### Official Review · Reviewer_Wtor · 2025-06-29

**Clarity:** 3
**Significance:** 2
**Originality:** 2
**Rating:** 4
**Confidence:** 3

**Summary:**

The paper presents Risk-Aware Direct Preference Optimisation (Ra-DPO), a new method for aligning LLMs with human preferences while taking risk into account. The method builds on token-level DPO (TDPO) and incorporates a nested risk measure from risk-sensitive reinforcement learning.
The authors want to include risk-awareness into LLM-training and take it into consideration, rather than simply maximising expected reward.
In the paper, risk is defined as the deviation (KL-divergence) from the original reference policy. The proposed loss function aims to maximise the likelihood of preferred responses while simultaneously penalising this token-level model drift.
The authors provide theoretical justification for their approach, deriving the loss function from a risk-aware advantage maximisation problem.
They present results of applying their method on IMDb dataset and Anthropic HH dataset, and evaluate the resulting models using AlpacaEval. They argue Ra-DPO achieves a better balance between alignment performance (reward accuracy, win-rate) and KL-divergence compared to the baselines.

**Questions:**

- How were the hyperparameters for the LLM experiments chosen?
- Could you run the LLM experiments on across a few more hyperparameter values? (beta, learning rate)
- Could you include performance of DPO (maybe with several different betas) in Figure 2?
- i think the paper could be improved by maybe using an external "risk evaluation" model to score risk rather than using drift from the reference model

**Ethical Concerns:**

["NO or VERY MINOR ethics concerns only"]

**Final Justification:**

The reviewers did address some of my concerns regarding the use of the word "risk" and put it into the context of the existing literature (SimPO vs DPO). However, I'm still not entirely convinced by the relatively minor performance gains and I find the title misleading

**Limitations:**

Yes

**Quality:**

2

**Strengths And Weaknesses:**

STRENGTHS:
- Paper is clear, easy to read, well structured
- The paper correctly identifies a limitation in current methods (DPO) as they are risk neutral (simply optimise for mean reward) and that TDPO, while moving to a more granular token-level approach, remains risk neutral. Proposing a risk-aware alternative method makes sense
- The structure of the experiments is sound, training on IMDb, Anthropic and evaluating on AlpacaEval and Ra-DPO shows marginal improvements on the baseline (in AlpacaEval). In Figure 3 they also show improved reward accuracy and KL-divergence from the reference model compared to TDPO. I would have been interested to see the performance of DPO here too

WEAKNESSES:
- I fundamentally disagree with the authors' framing of "risk" as divergence from the reference policy. The whole point of preference optimisation is to make the model less harmful, more helpful and honest. Quite a few papers have shown the KL divergence is useful to keep the model from moving completely out of distribution rather than to maintain any good qualities [1]. A reference model is not inherently "safe" or "correct". Constraining the policy to stay close to it can limit the model's ability to learn and adapt to the preferences it is being trained on. So much that recent, high-performing offline preference optimisation loss functions (such as SimPO [2]) completely scrap the KL-divergence term and instead just add an SFT term to the loss. This also has memory advantages (as you don't have to keep a copy of the model weights in memory)
- The paper's motivation that "a significant deviation from the reference model typically implies the degradation of superior decision-making and reasoning capabilities" is a strong claim presented without evidence.
- This narrow definition of risk is acknowledged, but only in the appendix
- Unconvincing Empirical Results: the improvements are marginal and no extensive hyperparameter search is performed. I understand the computational constraints, but given the marginal improvements I don't think the current evaluation is enough
- beta of 0.1 is quite low for DPO I think? I've seen 0.5 being used usually

[1] Is DPO Superior to PPO for LLM Alignment? A Comprehensive Study - Shusheng Xu, Wei Fu, Jiaxuan Gao, Wenjie Ye, Weilin Liu, Zhiyu Mei, Guangju Wang, Chao Yu, Yi Wu
[2] SimPO: Simple Preference Optimization with a Reference-Free Reward - Yu Meng, Mengzhou Xia, Danqi Chen

---

> ### Author Rebuttal · Authors · 2025-07-30
>
> We sincerely appreciate the valuable comments from the reviewer. We hope our responses below provide further clarity.
> #  W1: I fundamentally ……  in memory).
> **Response:** Thank you for the valuable suggestions. Below, we re-clarify the rationale behind defining "risk" as divergence from the reference policy and provide an explanation of the SimPO [1] compared to the DPO.
> ## Regarding the define of "risk":
> Indeed, as the reviewer pointed out, the core point of preference optimization is to make the model less harmful, more helpful, and truthful. However, drawing upon existing literature and our own insights, we would like to emphasize:
>
> - **A trade-off between alignment objectives and model fidelity is still necessary,** although the original(reference) model may not be "safe" or "correct". For example, in LLMs safety alignment tasks, a simple objective is to enable the model to reject unsafe responses while preserving original reasoning capabilities [2-3]. A response that is safe but logically incoherent or semantically uninformative is of little practical value. Therefore, many studies [4-5] typically formulate such tasks as constrained reward maximization problems.
> - **KL divergence has typically been used to penalize excessive deviations from a reference (critic) model [6-7].** In fact, numerous studies have reported that KL constraint offers many beneficial effects, such as balancing exploration and exploitation, ensuring stability and robustness, preventing catastrophic forgetting, and preserving the model’s fundamental capabilities [8-9].
>
> ## Regarding SimPO:
> SimPO [1] is a reference-free preference optimization algorithm that not only achieves superior performance, but also significantly reduces memory consumption. However, there are also studies indicating that SimPO has the following drawbacks:
> - As pointed out in [10], **the lack of a reference model in SimPO reduces training robustness and necessitates stricter conditions to prevent catastrophic forgetting.** The experimental results in the paper (Table 1) show that SimPO does not consistently outperform DPO. Specifically, SimPO demonstrates a significant advantage over DPO when training the Qwen2.5-7B-Base model, while DPO slightly outperforms SimPO when training the Qwen2.5-7B-Instruct model.
> - As noted in [11], different from DPO, **SimPO reintroduces complexity through dual parameters $(\beta, \gamma)$,** which introduce additional complexity on hyperparameter tuning.
> - As pointed out in Subsection 3.8.2 of [12], **the reward function in SimPO emphasizes length-normalized next token prediction, which raises concerns about a potential deviation from the original alignment objective.**
>
> **From our perspective,** DPO and SimPO serve as representative examples of reference-based and reference-free preference optimization methods, respectively. A comprehensive comparison in terms of performance, stability and robustness, hyperparameter tuning complexity, and computational efficiency reveals that each approach has its own trade-offs. Consequently, they are suited to different applications and both continue to represent valuable research directions.
>
> # W2: Regarding the evidence of our paper's motivation.
> **Response:** We have already provided some supporting evidence in our response to W1. Below, we present additional direct evidence:
> - As pointed out in [7], **in stylistic text continuation tasks, optimizing reward without constraints leads to incoherent continuations,** whereas introducing a KL constraint encourages the policy to remain close to a language model trained on BookCorpus.
> - As noted in [13], TDPO outperforms DPO while maintaining a lower KL divergence. Furthermore, we implemented our Ra-DPO and reproduced both TDPO and DPO baselines. As shown in the sampling results from AlpacaEval (see Supplementary Material, directory: "LLMs_eval_results / Q&A_evaluation_examples"), **DPO (with higher KL) tends to generate longer responses in many question-answering tasks, some of which contain several repetitive statements and lack logical coherence.** In contrast, such issues are rarely observed with TDPO and Ra-DPO.
> - **Several of our unsuccessful experiments suggest that model instability or collapse is frequently associated with increased KL divergence.** For instance, when setting $\alpha=0.1$, both TDPO2 and Ra-DPO show significantly elevated KL divergence, leading to poor performance on the AlpacaEval.
>
> # W3: Regarding the definition of risk.
> **Response:** In the new version, we will appropriately incorporate the explanation of risks into the main text.
>
> # W4-5 & Q1-3: Regarding the hyperparameters chosen and search.
> **Response: Hyperparameters chosen:** As mentioned in Section 4, we set the hyperparameters following the original KTO [14] implementation. Specifically, in the official KTO codebase, the default value of $\beta=0.1$ can be found in the file “HALOs/config/loss/dpo.yaml”, and the default value of $lr=5.0e^{-6}$ is specified in the file “HALOs/config/config.yaml”. This configuration is also widely adopted in other literature, including in implementations of DPO (Hugging Face version) [5] and TDPO [13].
>
> **Hyperparameter search:** We have conducted extensive hyperparameter searches for various algorithms, including DPO, TDPO2, and Ra-DPO2, across different settings of $\beta$ and learning rate (lr).
> However, due to the NeurIPS 2025 policy prohibiting the inclusion of  external links or PDF in the rebuttal, we summarize the key results in the tables below.
>
> **Table 1. Performance of various algorithms under different $\beta$ settings.** (Anthropic HH, Pythia-1.4B, H100, $lr=5.0e^{-6}$)
> | Model | | $\beta=0.1$ | | | $\beta=0.3$ | | | $\beta=0.5$ | |
> |-|-|-|-|-|-|-|-|-|-|
> | | Reward Accuracy | $D_{SeqKL}$ | Sharpe Ratio | Reward Accuracy | $D_{SeqKL}$ | Sharpe Ratio | Reward Accuracy | $D_{SeqKL}$ | Sharpe Ratio |
> | DPO| 0.51 | 9.61 | -2.86 | 0.52 | 1.05 | -0.95 | 0.51| 0.73 | -0.78 |
> | TDPO2 | 0.51 | 0.61 | 0.08 | 0.52 | 0.49 | 0.12 | 0.53 | 0.45 | 0.20 |
> | Ra-DPO2(CVaR) | 0.55 | 0.44 | 0.92 | 0.52 | 0.41 | 2.78 | 0.53 | 0.39 | 0.69 |
> | Ra-DPO2(ERM)  | 0.55 | 0.42 | 0.95 | 0.54 | 0.37 | 0.88 | 0.54 | 0.36 | 0.91 |
>
> **Table 2. Performance of various algorithms under different learning rates.** (Anthropic HH, Pythia-1.4B, A100, $\beta= 0.1$)
> | Model | | $lr=1.0e^{-6}$ | | | $lr=5.0e^{-6}$ | | | $lr=1.0e^{-7}$ | |
> |-|-|-|-|-|-|-|-|-|-|
> | | Reward Accuracy | $D_{SeqKL}$ | Sharpe Ratio | Reward Accuracy | $D_{SeqKL}$ | Sharpe Ratio | Reward Accuracy | $D_{SeqKL}$ | Sharpe Ratio |
> | DPO | 0.51 | 0.32  | -0.23 | 0.51 | 15.5 | -3.10 | 0.49 | 0.20 | 0.01 |
> | TDPO2 | 0.51 | 0.26 | 0.34 | 0.53 | 0.61 | 0.05 | 0.49 | 0.20 | 0.39 |
> | Ra-DPO2(CVaR) | 0.53 | 0.23 | 1.03 | 0.57 | 0.50 | 0.84 | 0.50 | 0.20 | 0.91|
> | Ra-DPO2(ERM) | 0.54 | 0.23 | 0.99 | 0.58 | 0.43 | 0.87 | 0.50  | 0.20 | 1.13 |
>
> **Note:**
> 1. Sharpe ratio [15, 16], a new evaluation metric, is introduced. A higher Sharpe ratio indicates that a model attains a given performance gain with lower behavioral deviation.
> 2. The data in the tables represent the average over the final stable phase of training. It is worth noting that DPO's KL divergence and Sharpe ratio are often unstable.
> 3. The experiments in Table 1 were conducted on two H100 GPUs, each with 80GB of GPU memory, while  the experiments in Table 2 were conducted on two A100 GPUs, each with 40GB of GPU memory.
>
> # Q4: Regarding the "risk evaluation" of model.
> **Response:** Thank you for the valuable suggestions. Measuring the divergence between reference model and policy model is a common practice [3, 5, 9, 13]. Introducing an external "risk evaluation" model could be a feasible direction for exploration; however, several foreseeable challenges may arise. For instance, during the initial training phase, the distribution of the trained model may differ significantly from that of reference model, which could destabilize or even collapse the training process. Selecting an appropriate reference model would thus be a critical challenge. In our future work, we will conduct further experiments in this direction.
>
> It is note that in our response to Reviewer mp5k, we provide detailed results on policy entropy, Sharpe ratio, and KL divergence across different models (including Pythia-1.4B and DeepSeek-R1-Distill-Qwen-1.5B), which may offer additional insights into this issue.
>
> # Reference:
> 1. Simpo: Simple preference optimization with a reference-free reward. NeurIPS, 2024.
> 2. Safe RLHF: Safe Reinforcement Learning from Human Feedback. ICLR, 2024.
> 3. Stepwise alignment for constrained language model policy optimization. NeurIPS, 2024.
> 4. Training a helpful and harmless assistant with reinforcement learning from human feedback. arXiv preprint arXiv:2204.05862, 2022.
> 5. Direct preference optimization: Your language model is secretly a reward model. NeurIPS, 2023.
> 6. Trust region policy optimization. ICML, 2015.
> 7. Fine-tuning language models from human preferences. arXiv preprint arXiv:1909.08593, 2019.
> 8. Overcoming catastrophic forgetting in neural networks. Proceedings of the national academy of sciences, 2017, 114(13): 3521-3526.
> 9. Beyond reverse KL: Generalizing direct preference optimization with diverse divergence constraints. ICLR, 2024.
> 10. Pre-dpo: Improving data utilization in direct preference optimization using a guiding reference model. arXiv preprint arXiv:2504.15843, 2025.
> 11. Repo: Relu-based preference optimization. arXiv preprint arXiv:2503.07426, 2025.
> 12. A comprehensive survey of llm alignment techniques: Rlhf, rlaif, ppo, dpo and more. arXiv preprint arXiv:2407.16216, 2024.
> 13. Token-level direct preference optimization. ICML, 2024.
> 14. Model alignment as prospect theoretic optimization. ICML, 2024.
> 15. The sharpe ratio. Streetwise–the Best of the Journal of Portfolio Management, 1998, 3(3): 169-85.
> 16. Adjusting for risk: An improved Sharpe ratio. International review of economics & finance, 2000, 9(3): 209-222.

---

> > ### Comment · Reviewer_Wtor · 2025-08-05
> >
> > Thank you for the clarifications, I've updated my score.

---

> > > ### Author Response · Authors · 2025-08-06
> > >
> > > Thanks for your advices and the score revision! We're very happy to have addressed your concerns and will make sure to include the corresponding revisions in a future version of the paper!

---

### Official Review · Reviewer_wuGU · 2025-07-03

**Clarity:** 3
**Significance:** 2
**Originality:** 2
**Rating:** 4
**Confidence:** 3

**Summary:**

This paper proposes Risk-aware Direct Preference Optimization (Ra-DPO), a LLM alignment method that includes risk-awareness by using a class of nested risk measures. The main idea of Ra-DPO is to incorporate the nested risk measure such as Conditional Value-at-Risk (CVaR) and Entropic Risk Measure (ERM) into Token-level Direct Preference Optimization (TDPO) to avoid the model drift from the reference model. This paper evaluates Ra-DPO on IMDb, Anthropic HH, and AlpacaEval. The experiment results show that Ra-DPO can improve the alignment performance, while avoiding the model drift.

**Questions:**

- Q1. How does Ra-DPO compare to the other method such as DPO and TDPO in terms of computational complexity?
- Q2. Is there any trade-off in incorporating risk-awareness into DPO or TDPO?
- Q3. In Figure 3, why does Ra-DPO with ERM work worse than Ra-DPO with CVaR?

**Ethical Concerns:**

["NO or VERY MINOR ethics concerns only"]

**Final Justification:**

I maintain my initial rating, 4: Borderline accept. The main idea of Ra-DPO is to incorporate the nested risk measure into Token-level Direct Preference Optimization (TDPO). In my initial review, I pointed out three weak points and three questions: W1. an extension of TDPO, W2. increased computational complexity in training time, W3. model performance, Q1. comparison of computational complexity, Q2. trade-off in incorporating risk-awareness, Q3. comparison between Ra-DPO with ERM and Ra-DPO with CVaR. The authors provided thoughtful responses to my comments and questions. The response helped me understand this paper better.

**Limitations:**

The authors provide the limitations of their work in Section D.1 (i.e., Limitations) of Appendix. However, I am not sure that the limitation section can be in Appendix.

**Paper Formatting Concerns:**

This paper does not have any major formatting isseus.

**Quality:**

3

**Strengths And Weaknesses:**

The strengths of this paper can be summarized as follows:
- S1. The authors derive the loss function of Ra-DPO by using the thorough theoretical analysis.
- S2. Even though the evaluation benchmarks (IMDb and Anthropic HH) are relatively simple, the authors provide comprehensive experiment results on those datasets.

The weaknesses of this paper can be summarized as follows:
- W1. Even though this paper is theoretically strong, the proposed method can be considered as a direct extension of the previous work TDPO (Token-level Direct Preference Optimization, ICML 2024).
- W2. The loss function of Ra-DPO includes the term that calculates $D_{SeqRR}(x, y; \pi_{ref} | \pi_{\theta})$ over the generated tokens $z$. This increases the computational complexity during training time.
- W3. Even though Ra-DPO can effectively control model drift, the model performance does not seem outperforming. For example, in experiments on AlpacaEval (i.e., Table 1), the win-rate of DPO, TDPO_1, and Ra-DPO_1 are 52.1, 51.9, and 53.5, respectively.

---

> ### Author Rebuttal · Authors · 2025-07-30
>
> We sincerely appreciate the valuable comments from the reviewer. We hope our responses below provide further clarity.
> # W1: Even though …… (Token-level Direct Preference Optimization, ICML 2024).
> **Response:** As the reviewer points out, our method is an extension of TDPO. However, it is important to emphasize that we introduce nested risk measures to effectively enhance the risk sensitivity of the policy optimization process. **Our key theoretical contributions include:**
> - Incorporating nested risk measures into token-level policy optimization and providing a closed-form solution. Importantly, our method maintains a natural and simple loss function shown in Figure 1 (the sum of DPO loss and negative sequence risk ratio).
> - Establishing the connection between risk-aware state-action value functions and optimal policies. The key technical contributions lie in:
> - A risk-aware advantage function design under nested risk measures;
> - Proof of Bellman-type model equivalence with the Regret Preference Model under nested risk measures.
>
> **Main experiments results include:**
> - In the paper, we evaluate the effectiveness of the proposed method across various text generation tasks and analyze its sensitivity to the risk control parameter. The results show that our method effectively suppresses the risk of undesirable behavior while improving overall performance.
> - To further demonstrate the algorithm's properties, **we provide additional analyses on policy entropy, Sharpe ratio [1–2], hyperparameter sensitivity, and other key aspects for each baseline.** These can be found in Table 4 and in responses to the other reviewers. The results show that models trained with Ra-DPO2 achieve high reward accuracy, preserve policy entropy (indicating sustained exploration), attain a high Sharpe ratio (reflecting superior risk-adjusted performance), and maintain low KL divergence (ensuring proximity to the reference policy), highlighting the robustness and broad applicability of our approach.
>
> # W2 & Q1: Regarding the computational complexity during training time.
> **Response:** Thank you for the valuable suggestions. Before analyzing computational complexity, it is worth noting that we reformulate the recursive Bellman equation into a classical Bellman equation by constructing an augmented-state MDP despite introducing nested risk measures. As a result, **in terms of computational complexity, the main difference between our algorithm and TDPO lies in the use of risk measures such as CVaR or ERM, as opposed to the expectation.**  Below, we present their time and space complexity.
>
> **Table 1. Complexity of different risk measures.**
>
> | Method | Time Complexity | Space Complexity |
> |-|-|-|
> | Expectation | $O(B \cdot T)$ | $O(B \cdot T)$ |
> | ERM | $O(B \cdot T \cdot N)$ | $O(B \cdot T \cdot N)$ |
> | CVaR | $O(B \cdot T \cdot N \log N)$ | $O(B \cdot T \cdot N)$ |
>
> For a more intuitive presentation, we present performance and runtime of various algorithms on A100 and H100 GPUs under different models and datasets in the following tables.
>
> **Table 2. Runtime of various algorithms.** (IMDB, A100)
>
> | Model | DPO	| TDPO2 | Ra-DPO2(ERM) | Ra-DPO2(CvaR)|
> |-|-|-|-|-|
> | Gpt2l | 77min	| 80min | 82min | 83min|
>
> **Table 3. Runtime of various algorithms.** (Anthropic HH, A100)
>
> | Model | DPO | TDPO2 | Ra-DPO2(ERM) | Ra-DPO2(CvaR)
> |-|-|-|-|-|
> | Pythia1.4 | 262min | 271min | 356min | 369min |
> | Pythia2.8 | 13.7h | 13.9h | 14.3h | 15.2h |
>
> **Table 4.  Performance and runtime of various algorithms.** (Anthropic HH, Pythia-1.4B, H100, $\beta=0.1, lr=5.0e^{-6}$)
>
> | Model | Reward Accuracy | $D_{SeqKL}$ | Sharpe Ratio | Policy Entropy | Runtime |
> |-|-|-|-|-|-|
> | DPO | 0.51 | 9.61 | -2.86 | [2.93→2.82] | 179min |
> | TDPO2 | 0.51 | 0.61 | 0.08 | [2.93→2.89] | 185min |
> | Ra-DPO2(CVaR) | 0.55	| 0.44 | 0.92 | [2.93→2.91] | 207min |
> | Ra-DPO2(CvaR-v2) | 0.54 | 0.45 | 0.83 | [2.93→2.91] | 196min |
> | Ra-DPO2(ERM) | 0.55 | 0.42 | 0.95 | [2.93→2.91] | 192min |
>
> **Experimental analysis:**
> - Several variants of Ra-DPO consistently achieve high reward accuracy and Sharpe ratios, while maintaining low KL divergence. Notably, these improvements are achieved without a significant increase in training runtime.
>
> **Note:**
> 1. In Table 1, B denotes the batch size, T denotes the sequence length, and N denotes the number of samples per distribution.
> 2. The Sharpe ratio is a widely adopted metric for assessing risk-adjusted performance. In the context of language model alignment, we adopt it to assess the risk-adjusted reward. A higher Sharpe ratio indicates that a model attains a given performance gain with lower behavioral deviation—thus reflecting more stable and reliable training dynamics.
> 3. The runtime reported in Tables 2 and 3 was measured on two A100 GPUs, each with 40 GB of memory, while the runtime in Table 4 was measured on two H100 GPUs, each with 80 GB of memory.
> 4. Ra-DPO2 (CvaR-v2) is another version of Ra-DPO2 (CvaR), which effectively alleviates computational pressure by using top-k instead of quantile when calculating CVaR. Notably, our experiments show that the impact of this substitution on metrics such as reward accuracy and KL divergence is negligible.
>
> # W3: Even though Ra-DPO can…… and 53.5, respectively.
> **Response:** We acknowledge that the win rate of our method on AlpacaEval is not significantly higher than that of the baselines. However, it is important to note that the core idea of our approach lies in incorporating risk-awareness by employing a class of nested risk measures, which achieve a better balance between alignment performance and model drift under risky conditions. The token-level sequential risk ratio constraint makes the model more conservative in exploration when facing high-risk situations, leading to generated responses that are shorter and more moderate in length. As a result, our method achieves better performance in terms of length-controlled win rate as show in Table 1.
>
> Moreover, **there are several experimental results are worth highlighting:**
> - Figure 13 in Appendix C.4 presents results on the Anthropic HH dataset using Pythia-1.4B as the base model. For all algorithms, we report the average performance across three random seeds, with the shaded area indicating one standard deviation around the mean. The figure shows that, under both the CVaR-based and the ERM-based risk measure with µ = 5, the proposed algorithm achieves reward accuracy comparable to that of the baseline methods, but with greater stability, while maintaining consistently low sequential KL divergence.
> - In our responses to the other reviewers, we provided results on policy entropy, hyperparameter search, and other aspects for each algorithm. The results show that Ra-DPO (ERM) exhibits the most desirable risk-aware behavior among the evaluated methods, as evidenced by its high and stable Sharpe Ratio, as well as well-preserved policy entropy.
> - In the Supplementary Material under the directory " Q&A evaluation examples using LLMs\3-Evaluation results of LLMs", we present a comparative evaluation of our algorithm and baseline methods using DeepSeek and GPT-4o, based on a selected set of questions. The evaluation dimensions include:
>     - **Riskiness:** Whether the answer has potential risks or problems.
>     - **Effectiveness:** Whether the answer can effectively solve the problem.
>     - **Relevance:** Whether the answer closely revolves around the core of the question.
>     - **Redundancy:** Whether the answer has unnecessary repetitions or redundant information.
>
> The evaluation results show that our algorithm (Ra-DPO) performs well in terms of riskiness and relevance, but there are differences in effectiveness and redundancy.
>
> **The original conclusion is as follows:**
> - Some algorithms (such as Ra-DPO1 (CVaR, $\alpha=0.5, \mu=0.97$)  and Ra-DPO2 (CVaR, $\alpha=0.5, \mu=0.97$)  can provide high-quality practical information, while the effectiveness of other algorithms (such as Ra-DPO1 (CVaR, $\alpha=0.5, \mu=0.99$) ) and Ra-DPO2 (CVaR, $\alpha=0.5, \mu=0.99$) ) is relatively low and their redundancy is relatively high.
> - Future research should focus on optimizing algorithms to improve their performance in terms of effectiveness and redundancy, thereby enhancing their overall performance.
>
> # Q2. Is there any trade-off in incorporating risk-awareness into DPO or TDPO?
> **Response:**  There exists a certain trade-off when incorporating risk awareness into TDPO. As shown in the responses to W2 and Q1, Ra-TDPO2 achieves higher performance and maintaining lower KL divergence, while requiring slightly more running time.
>
> # Q3. In Figure 3, why does Ra-DPO with ERM work worse than Ra-DPO with CVaR?
> **Response:** This may be related to the inherent characteristics of the CVaR and ERM risk measures themselves. CVaR directly optimizes performance on the worst-case samples, whereas ERM applies exponential weighting uniformly across all samples, potentially diluting attention to extreme poor-performing cases. Additionally, ERM is more sensitive to hyperparameters, as its performance varies significantly with the choice of parameter. As shown in Figure 3, when the risk coefficient of Ra-DPO2 with ERM is set to $\mu=5$, it achieves performance comparable to that of Ra-DPO2 with CVaR.
>
> # Limitations: The authors provide …… can be in Appendix.
> **Response:** In the new version, we will move the discussion on limitations from the appendix to the main text.
>
> # Reference:
> 1. Sharpe W F. The sharpe ratio. Streetwise–the Best of the Journal of Portfolio Management, 1998, 3(3): 169-85.
> 2. Dowd K. Adjusting for risk: An improved Sharpe ratio. International review of economics & finance, 2000, 9(3): 209-222.

---

> > ### Comment · Reviewer_wuGU · 2025-08-08
> >
> > Thank you for providing thoughtful responses to my questions and comments. The responses helped me understand the paper better. Here are some thoughts on the responses:
> > - I appreciate that the authors provided additional experiments on training time of DPO, TDPO, and Ra-DPO. The experiment results show that training Pythia-2.8B on Anthropic HH with Ra-DPO increase the training time about 1.5 hour. This may not be considered as huge increase. However, if the model size and the training data size increase, the training time may increase considerably.
> > - I appreciate that the authors provided additional explanations on the experiment results worth to note.
> >
> > After carefully reviewing the author response, I maintain my initial rating.

---

> > > ### Author Response · Authors · 2025-08-08
> > >
> > > Thank you for reading our rebuttal and for the valuable suggestions. We sincerely appreciate your positive evaluation and the helpful discussions, which indeed helped us to improve the quality of this work.

---

### Official Review · Reviewer_mp5k · 2025-07-21

**Clarity:** 4
**Significance:** 3
**Originality:** 3
**Rating:** 5
**Confidence:** 3

**Summary:**

The authors highlight a couple issues with industry standard techniques:
1. The lack of granular token-level likelihood and optimization (issues faced by DPO and its variants).
2. The lack of consideration of the overall distribution (i.e. the tails) during the reward maximization approach for DPO variants that do not suffer from the issue highlighted in #1 (such as for TDPO, which only optimizes the expected reward/the mean). Many variants that do address this fail to address #1.

They are proposing a new method that hits #1 and #2, and have termed this Ra-DPO. They are proposing a new method that considers the shape of the reward distribution when maximizing the expected return. They do this by replacing the expectation by a nested risk functional which pushes the downstream reward return distribution through a coherent risk analysis (to maintain convexity), by focusing on the worst outcomes (to focus on the tails parameterized by mu) and by propagating that token by token (basically if any downstream token contains a high risk tail, the current token reflects it). The authors also proceed to show the efficacy of it through empirical trials.

The contribution are two new objectives (Ra-DPO1/2) that directly address both of the issues with current widely adopted optimization objectives, and the theory that underpins those assertions + empirical results for those two new objectives compared to the currently adopted objectives.

**Questions:**

1. How does the efficacy of this metric change with the base model's entropy? How does it compare to when the base model is very peaky on the data this is being applied to versus when it is more OOD to the base model? I think this is my main concern with this approach, since it relies quite heavily on the base model's entropy and the ability to sample the tail. An explicit section on this (with respect to having concrete metrics from the base model itself to understand its capability and behavior on the evaluated task) would be helpful.
2. How does the gradients and the E2E efficacy look on tasks with rewards that benefit from varying answer lengths? Some tasks require longer answers and others shorter, how does the efficacy of the methodology wrt the baselines shift on these tasks? I do not think the length controlled scores from AlpacaEval satisfactorily covers this, as the target of this question is the potential for noisy gradients as the length increases for tasks that demand it, while the AlpacaEval communicates better accuracy at lower generation lengths.
3. Can you probe Ra-DPO on benchmarks that explicitly measure the rare but severe failure modes and report more tail-focused metrics from these? Ideally, it would be good to have some way to qualify the shift in model's answers specifically made possible by capturing the entire reward distribution -- including the tail.

**Ethical Concerns:**

["NO or VERY MINOR ethics concerns only"]

**Final Justification:**

The authors and I discussed their baseline methodology. The authors mentioned wanting to provide additional information in a revised version of the paper pursuant to some of the provided comments, and provided references to existing experiments they have run for the rest of the comments. The primary issue around the rigor of the baselines still exist, but the additional information strengthens the paper further along the lines that the paper was already strong in. However, since my original rating was already high and -- in my opinion -- accurately captured all the strengths of the paper and the additional information the authors provided only reinforced those strengths, I maintained my rating (as the edits has not bumped it up to the next tier).

**Limitations:**

Broadly yes, but the rigor of the baselines is a limitation of the paper. The paper is good and the idea is novel, but the paper can be made stronger by having more practical feasibility studies of the approach. In general, being able to provide reasoned guidance about the the dependency on the reference entropy, the gradient noise response with respect to the generation length, and more tail-oriented studies would greatly strengthen the paper. It is possible to do this with limited compute by primarily having good coverage of base models and tasks along these axis in my opinion.

**Paper Formatting Concerns:**

No formatting errors.

**Quality:**

4

**Strengths And Weaknesses:**

Quality and Clarity: The quality of the paper is strong in my own opinion. I see a strong narrative whereby the authors start with drafting the limitations of the current methodologies in industry along two relevant axis (i.e. capturing the entire reward distribution and the need for token-wise maximization) and then directly address the limitations through a new formulation that is theoretically underpinned rigorously. The proofs are well written and is relatively easy to follow, and the limitations and workarounds are clearly indicated. I also appreciate the augmentations made to the formulations for mechanistic purposes, making it easier to adopt.

The practicality study of the methodology needs more work in my opinion to get a strong significance score. I fully acknowledge the three dataset (IMDB, Anthropic-HH, and AlpacaEval) + Two Model family (Pythia-XX,  GPT2) experiments, but they seem a bit limited due to how dated they are, and how saturated the results are.

The baselines are the most significant weakness of this paper, with IMDB (Figure 2) showing very saturated rewards across hyperparameters (almost impossible to discern impact). Figure 3 and Table 1 are more meaningful, but requires more model baselines. The varying in models is appreciated, but the authors do not sufficiently explore the connection of the base models to the task being studied despite relying so heavily on the reference policy entropy. I can conceive of a situation where the base policy is very peaky (due to heavy SFT, etc.) and the sampling of the tail being more difficult. This is especially true when the data being optimized is very in-distribution to the base reference policy SFT data. This relation does not seem to be comprehensively studied. It would be helpful to see how this approach practically translates to those situations, which are relatively common. Also, I don't see direct metrics that address their core claims, adding in more tail-sensitive practical metrics would be helpful here to directly address the claims that are made initially, instead of solely adding E2E numbers since they do claim that capturing the entire reward distribution is a big part of their contribution. Having concrete metrics that directly show it would be helpful. Overall strong paper, but this would improve it.

The idea of jointly optimizing for the reward distribution shape and a per-token reward maximization is fantastic. Even beyond safety, the tail is a critical part of the out of distribution quality, and ensuring minimal degradation with respect to the base policy is very meaningful (more so on verifiable and reasoning-benefitting tasks, which isn't covered here). Having a way of optimizing for both jointly is good contribution -- the only major limitation of the paper being the practicality and feasibility study comprehensiveness.

---

> ### Author Rebuttal · Authors · 2025-07-30
>
> We sincerely appreciate the valuable comments from the reviewer. We hope our responses below provide further clarity.
> # Q1 & Q3. Regarding the model's entropy, practical metrics and model baselines.
> **Response:** To more directly demonstrate the characteristics of our algorithm, we have added experiments with multiple metrics under different model baselines.
>
> **Model baselines:** We conducted experiments with Pythia-1.4B and DeepSeek-R1-Distill-Qwen-1.5B [1] on the Anthropic HH dataset.
>
> **Practical metrics:** We added policy entropy and Sharpe ratio [2-3] to evaluation our method. The Sharpe ratio is a widely adopted metric for assessing risk-adjusted performance. In the context of language model alignment, we adapt it to assess the risk-adjusted reward. A higher Sharpe Ratio indicates that a model attains a given performance gain with lower behavioral deviation—thus reflecting more stable and reliable training dynamics.
>
> However, due to the NeurIPS 2025 policy prohibiting the inclusion of external links or PDF in the rebuttal, we summarize the key results in the table below.
>
> **Table 1. Performance and runtime of various algorithms.** (Anthropic HH, Pythia-1.4B, H100, $\beta=0.1, lr=5.0e^{-6}$）
>
> | Model | Reward Accuracy | $D_{SeqKL}$ | Sharpe Ratio | Policy Entropy | Runtime |
> |-|-|-|-|-|-|
> | DPO | 0.51 | 9.61 | -2.86 | [2.93→2.82] | 179min |
> | TDPO2 | 0.51 | 0.61 | 0.08 | [2.93→2.89] | 185min |
> | Ra-DPO2(CVaR) | 0.55	| 0.44 | 0.92 | [2.93→2.91] | 207min |
> | Ra-DPO2(CvaR-v2) | 0.54 | 0.45 | 0.83 | [2.93→2.91] | 196min |
> | Ra-DPO2(ERM) | 0.55 | 0.42 | 0.95 | [2.93→2.91] | 192min |
>
> **Experimental results in Table 1:**
> - The model trained with DPO exhibits low reward accuracy, high KL divergence, and a negative Sharpe ratio, indicating poor alignment performance and unstable policy optimization.
> - Models trained using different Ra-DPO variants maintain a stable policy entropy of approximately 2.93 across the entire training process. In contrast, models trained with baseline methods show a consistent downward trend in entropy, persisting until the end of training.
> - **Several variants of Ra-DPO consistently achieve high reward accuracy and Sharpe ratios, while maintaining low KL divergence.** Notably, these improvements are achieved without a significant increase in training runtime.
>
> **Table 2. Performance and runtime of various algorithms.** (Anthropic HH, DeepSeek-R1-Distill-Qwen-1.5B, H100, $\beta=0.1, lr=5.0e^{-6}$)
> | Model | Reward Accuracy | $D_{SeqKL}$ | Sharpe Ratio | Policy Entropy | Runtime |
> |-|-|-|-|-|-|
> | DPO | 0.49 | 0.45 | -0.75 | [2.89→2.85] | 236min |
> | TDPO2 | 0.52 | 0.37 | -0.08 | [2.89→2.86] | 242min |
> | Ra-DPO2(CVaR) | 0.54	| 0.34 | 0.93 | [2.89→2.88] | 261min |
>
> **Experimental results in Table 2:**
> - Experiments conducted using the DeepSeek-R1-Distill-Qwen-1.5B model yield patterns similar to those reported in Table 1, validating the consistency of our findings.
> - A closer look at Table 2 reveals a key distinction: while DPO and TDPO2 exhibit marginally negative Sharpe ratios, indicating limited risk-adjusted performance, Ra-DPO continues to achieve a substantially higher Sharpe ratio. **This underscores the stability and generalizability of Ra-DPO under varying model backbones.**
>
> **Experimental analysis:**
> - A comprehensive analysis of Tables 1 and 2 reveals that Ra-DPO2 delivers consistently strong performance across two distinct model backbones: Pythia-1.4B and DeepSeek-R1-Distill-Qwen-1.5B. **Specifically, models trained with Ra-DPO2 attain high reward accuracy, preserve policy entropy (indicating sustained exploration), achieve a high Sharpe ratio (reflecting superior risk-adjusted performance), and maintain low KL divergence (ensuring proximity to the reference policy). These results highlight the robustness and broad applicability of our approach.**
>
> **Note:**
> 1. The values of reward accuracy, KL divergence, and Sharpe ratio reported in the Table 1 and 2 represent the average over the final stable phase of training, after convergence. The policy entropy of DPO and TDPO2 does not converge, but instead continues to decrease gradually throughout training.
> 2. The experiments in Tables 1 and 2 were conducted on two H100 GPUs, each with 80GB of GPU memory.
> 3. In our responses to the other reviewers, we provide additional analyses on complexity analysis, Sharpe ratio, hyperparameter sensitivity, and other key aspects for each baseline.
> 4. Ra-DPO2 (CvaR-v2) is another version of Ra-DPO2 (CvaR), which effectively alleviates computational pressure by using top-k instead of quantile when calculating CVaR. Notably, our experiments show that the impact of this substitution on metrics such as reward accuracy and KL divergence is negligible.
>
> # Q2. How does the …… generation lengths.
> **Response:** We appreciate the reviewer’s insightful comment on the complex interaction among answer length, reward shaping, gradient variance, and overall E2E performance. Indeed, disentangling these factors is non-trivial and remains an open challenge in RL-based language generation. Due to time constraints, a comprehensive analysis is currently infeasible. However, as shown in Tables 1 and 2 and our supplementary material, **several key points can be observed:**
> - **There exists a certain trade-off when incorporating risk awareness into TDPO.** As shown in Tables 1 and 2, Ra-TDPO2 attain high reward accuracy, preserve policy entropy, achieve a high Sharpe ratio, and maintain low KL divergence, while requiring slightly more running time.
> - While our method does not explicitly control for output length, we observe in the Supplementary Material that **models trained with Ra-DPO2 tend to generate more concise and focused responses without sacrificing reward,** suggesting a potential regularization effect.
>
> We will include a discussion on this point in the final version.
>
> ## Regarding Supplementary Material:
> - In the directory: "LLMs_eval_results / Q&A_evaluation_examples" in Supplementary Material, the sampling results from AlpacaEval show DPO (with higher KL) tends to generate longer responses in many question-answering tasks, some of which contain several repetitive statements and lack logical coherence. In contrast, such issues are rarely observed with TDPO and Ra-DPO.
>
> - In the directory " Q&A evaluation examples using LLMs\3-Evaluation results of LLMs" in the Supplementary Material, we present a comparative evaluation of our algorithm and baseline methods using DeepSeek and GPT-4o, based on a selected set of questions. The evaluation dimensions include:
>     - **Riskiness:** Whether the answer has potential risks or problems.
>     - **Effectiveness:** Whether the answer can effectively solve the problem.
>     - **Relevance:** Whether the answer closely revolves around the core of the question.
>     - **Redundancy:** Whether the answer has unnecessary repetitions or redundant information.
>
> The evaluation results show that our algorithm (Ra-DPO) performs well in terms of riskiness and relevance, but there are differences in effectiveness and redundancy.
>
>
> #  Reference:
> 1. Guo D, Yang D, Zhang H, et al. Deepseek-r1: Incentivizing reasoning capability in llms via reinforcement learning. arXiv preprint arXiv:2501.12948, 2025.
> 2. Sharpe W F. The sharpe ratio. Streetwise–the Best of the Journal of Portfolio Management, 1998, 3(3): 169-85.
> 3. Dowd K. Adjusting for risk: An improved Sharpe ratio. International review of economics & finance, 2000, 9(3): 209-222.

---

> ### Comment · Reviewer_mp5k · 2025-08-06
> **Acknoledgement**
>
> Thank you for the clarifications. I think adding in the additional discussions around these will allow the final version to be stronger. I will retain my current score since that is primarily underscored by the rigor of the benchmarks chosen for the points raised and my current score accurately provides the metrics for this great paper, but otherwise, I am aligned with the discussion items.

---

> > ### Author Response · Authors · 2025-08-06
> >
> > Thank you for your recognition and valuable feedback. We will further improve our work based on your previous suggestions and present the revisions in the updated version.

---

### Note · Authors · 2025-08-12

We thank the reviewers and the AC for their valuable feedback and for their efforts in enhancing the quality of the paper. Following the rebuttal phase, all reviewers have acknowledged our clarifications regarding the methodological innovation, theoretical contributions, and experimental validation, and have indicated that they are maintaining or increasing their scores.

Regarding the concern raised by Reviewer wuGU about "training time", we have proposed mitigation strategies, such as using top-k selection instead of quantile computation in the CVaR calculation. Notably, our experiments show that the impact of this substitution on metrics such as reward accuracy and KL divergence is negligible.

In the final version, we will make the following key revisions based on the rebuttal discussion:
- In the main text, we will provide a clearer exposition of the risk definition and our contributions, make appropriate adjustments to the experiments, and move the analysis of the paper’s limitations to the end;
- In the appendix, we will include an analysis of computational complexity as well as additional experimental results and discussions.

---

### Decision · Program_Chairs · 2025-09-17

**Decision:**

Accept (poster)

**Comment:**

The paper introduces Risk-aware direct preference optimization (RA-DPO), a novel LLM alignment method that incorporates risk-awareness with a class of nested risk measures. The authors addressed most of the concerns during the rebuttal, and all reviewers are positive about the paper.